# In silico analysis of the transcriptional regulatory logic of neuronal identity specification throughout the *C. elegans* nervous system

Lori Glenwinkel[1], Seth R Taylor[2], Kasper Langebeck-Jensen[3], Laura Pereira[1], Molly B Reilly[1], Manasa Basavaraju[4,5], Ibnul Rafi[1], Eviatar Yemini[1], Roger Pocock[3,6], Nenad Sestan[4,5], Marc Hammarlund[4,5], David M Miller III[2], Oliver Hobert[1]*

[1]Department of Biological Sciences, Columbia University, Howard Hughes Medical Institute, New York, United States; [2]Department of Cell and Developmental Biology, Vanderbilt University School of Medicine, Nashville, United States; [3]Biotech Research and Innovation Centre, University of Copenhagen, Copenhagen, Denmark; [4]Department of Neurobiology, Yale University School of Medicine, New Haven, United States; [5]Department of Genetics, Yale University School of Medicine, New Haven, United States; [6]Development and Stem Cells Program, Monash Biomedicine Discovery Institute and Department of Anatomy and Developmental Biology, Monash University, Melbourne, Australia

**Abstract** The generation of the enormous diversity of neuronal cell types in a differentiating nervous system entails the activation of neuron type-specific gene batteries. To examine the regulatory logic that controls the expression of neuron type-specific gene batteries, we interrogate single cell expression profiles of all 118 neuron classes of the *Caenorhabditis elegans* nervous system for the presence of DNA binding motifs of 136 neuronally expressed *C. elegans* transcription factors. Using a phylogenetic footprinting pipeline, we identify *cis*-regulatory motif enrichments among neuron class-specific gene batteries and we identify cognate transcription factors for 117 of the 118 neuron classes. In addition to predicting novel regulators of neuronal identities, our nervous system-wide analysis at single cell resolution supports the hypothesis that many transcription factors directly co-regulate the cohort of effector genes that define a neuron type, thereby corroborating the concept of so-called terminal selectors of neuronal identity. Our analysis provides a blueprint for how individual components of an entire nervous system are genetically specified.

*For correspondence:
or38@columbia.edu

## Introduction

Nervous systems are composed of an enormous diversity of neuronal cell types. Traditionally, in both vertebrates and invertebrates, neuronal cell types were characterized solely based on anatomical features (*Bota and Swanson, 2007*; *Cajal, 1911*; *White et al., 1986*). With the advent of molecular biology and, more recently, single cell transcriptome profiling, neuronal cell type classification has begun to increasingly rely on molecular features (*Hobert et al., 2016*; *Tasic, 2018*; *Zeng and Sanes, 2017*). As one might predict, neuronal cell type identity and diversity are defined by neuron type-specific gene batteries that encode the specific structural and functional features of a mature neuron type. While each neuron type expresses a unique gene battery, each individual genetic

component of a gene battery is usually expressed in more than one neuron type. For example, the biosynthetic enzymes and transporters that define a specific neurotransmitter phenotype are expressed by multiple distinct cell types (e.g. acetylcholine is used as a neurotransmitter by spinal cord motor neurons, as well as projection neurons in the basal forebrain). Hence, neuron-specific gene batteries are defined by unique combinations of not uniquely expressed genes.

To understand how cell types in a nervous system differentiate into many distinct neuron types, one needs to understand how the expression of neuron type-specific gene batteries are controlled. Even fully mature neurons generally express many transcription factors (TFs) and one could therefore envision two distinct, extreme models for how neuron type-specific gene batteries are activated: In one case, the 'load' of controlling gene expression during terminal neuronal differentiation may be distributed somewhat evenly over many distinct TFs, each of which controls expression of subsets of genes of a neuron type-specific gene battery (*Figure 1A*). Contrasting such a 'piecemeal' regulatory model, one could also envision that among the cohorts of TFs expressed in a terminally differentiating neuron type, some TFs may have a broader role than others. In the most stringent version of this model, a 'coordinated regulatory' model, a TF (or a cohort of multiple TFs) may jointly regulate most if not all genes of the gene battery that uniquely differentiates one neuron type from another neuron type (*Figure 1A*).

These two models have been addressed to some extent in the nematode *Caenorhabditis elegans*, which offers a number of key advantages for studying this problem. Its nervous system is composed of only 302 neurons, which fall into a wide range of 118 anatomically defined and molecularly distinct neuron classes (*Hobert et al., 2016*; *White et al., 1986*). For each anatomically defined neuron class, a wealth of reporter transgene-based expression data has been available, both for TFs and potential effector gene batteries, that is, genes that encode the 'nuts and bolts' of a mature neuron type, such as neurotransmitter receptors, ion channels, neuropeptides, and others (*Hobert, 2016b*). These molecular markers have been instrumental for genetic mutant analysis that has identified several distinct TFs that are required for the execution of the differentiation programs of distinct neuron types. The first paradigm of such an analysis involved the two homeobox genes *unc-86* and *mec-3*, identified by their effect on the differentiation of touch sensory neurons (*Chalfie et al., 1981*; *Way and Chalfie, 1988*). Both TFs were found to heterodimerize to directly control the expression of scores of reporter genes that define the phenotype of differentiated touch sensory neurons (*Duggan et al., 1998*; *Xue et al., 1993*; *Zhang et al., 2002*). Four additional cases corroborated the UNC-86/MEC-3 scenario and are schematized in *Figure 1B*. Here again, specific TFs (or heterodimeric combinations thereof) were found to directly control the expression of a large cohort of terminal effector genes in specific neuron classes (*Etchberger et al., 2007*; *Kratsios et al., 2011*; *Masoudi et al., 2018*; *Wenick and Hobert, 2004*). In all cases, nearly all molecular markers that characterize a specific neuron type fail to be expressed in animals lacking the respective TF(s). Moreover, regulation of these molecular markers is often direct, as inferred by (a) the identification of *cis*-regulatory binding sites for all of these individual TFs, (b) the presence of these binding sites in effector genes, and (c) the experimentally verified importance of these *cis*-regulatory motifs for gene expression in the respective neuron types (*Figure 1C*). The continuous expression of these factors throughout the life of the respective neuron types combined with conditional gene removal has further illustrated that these factors are not only required for initiation of the respective neuron type-specific gene batteries but also continuously required to maintain their expression. Taken together, these findings strongly suggest that individual TFs define the differentiated state of a neuron through a coordinated regulatory mechanism (*Figure 1A*). These TFs were called 'terminal selectors' (*Hobert, 2008*; *Hobert, 2016b*).

One key question that has remained is how generalizable the concept of coordinated transcriptional control by terminal selector TFs really is across the entire *C. elegans* nervous system. A number of studies have attempted to address this question. Based on the analysis of small number of molecular markers in TF mutant backgrounds, multiple TFs have emerged as candidate terminal selectors (summarized in *Hobert, 2016a*). For example, in the context of studying glutamatergic neurotransmitter identity specification, we identified a number of TFs that control *eat-4/VGLUT* expression and we showed that these factors also control a small number of additional identity features (*Serrano-Saiz et al., 2013*). However, the often very limited number of molecular markers tested in the TF mutant backgrounds (*Figure 1D*) has left the question open whether these TFs are indeed broadly controlling the many distinct identity features of a neuron. Moreover, with the

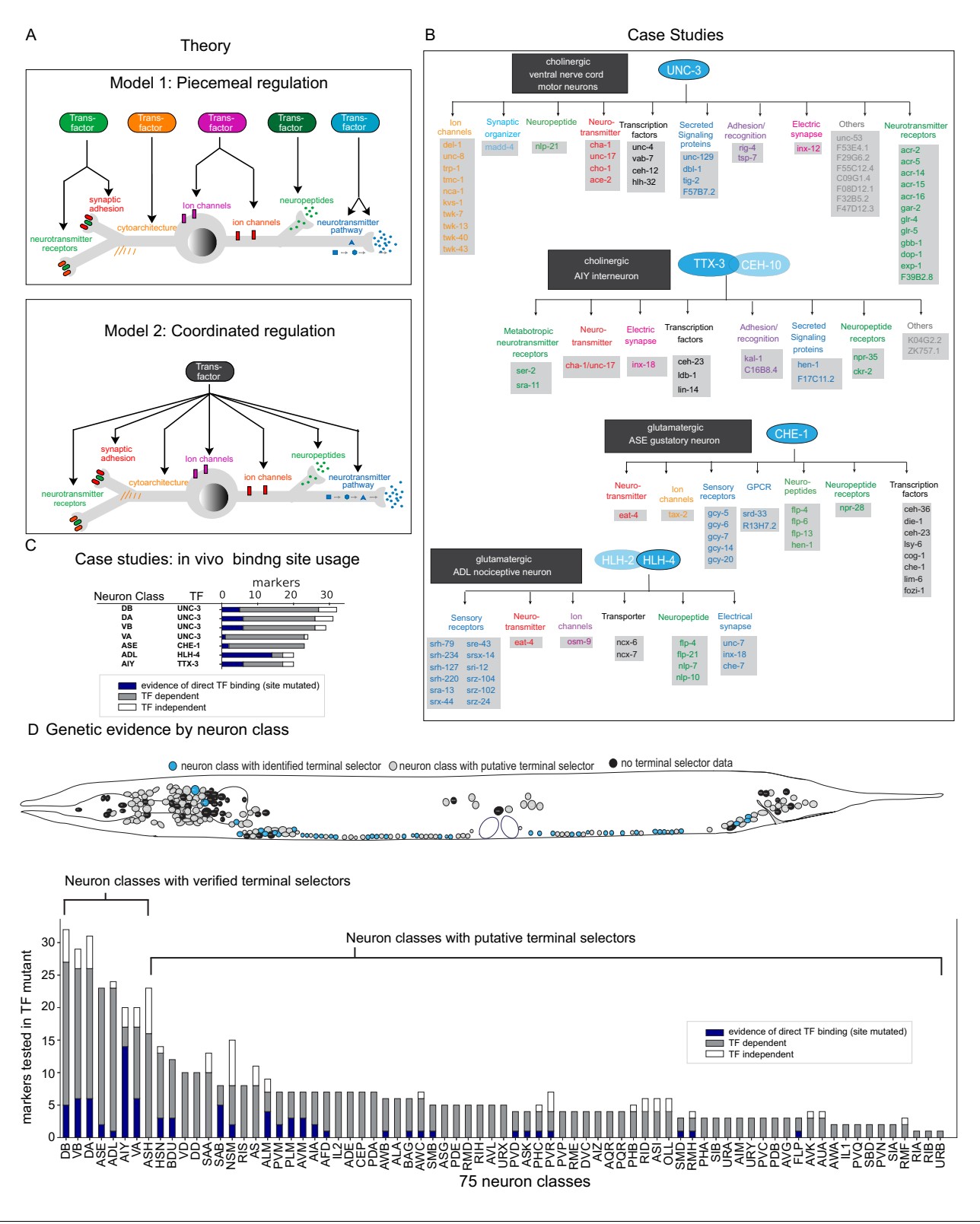

**Figure 1.** Background. (**A**) Possible models for regulation of terminal gene batteries. (**B**) Four examples of terminal selectors (>20 markers tested >1 binding site mutated). All genes shown are direct targets of the indicated terminal selectors (blue ovals) as evidenced by loss of reporter expression in genetic loss-of-function mutants (*Etchberger et al., 2007*; *Kratsios et al., 2011*; *Masoudi et al., 2018*; *Wenick and Hobert, 2004*). (**C**) Genetic and biochemical evidence of direct regulation by four terminal selectors. Blue: TF binding site mutated resulting in loss of reporter gene expression. Gray:

*Figure 1 continued on next page*

*Figure 1 continued*

reporter gene requires TF for expression in neuron class. White, reporter gene expression not affected in TF mutant. (D) Summary of published marker tested in putative terminal selector mutants per neuron class selectors, putative terminal selectors = other classes with genetic evidence of regulation (see *Hobert, 2016a* for review of markers tested). Neuron classes with more than one putative terminal selector list the largest set of markers tested for a single TF.

notable exception of the cases mentioned above (and shown in *Figure 1B–D*), it is not clear whether most of these TFs *directly* control terminal effector genes and, hence, whether they are indeed the best candidate terminal selectors for the given neuron classes. Lastly, the number of known transcriptional regulators of neuronal identity remains limited (*Hobert, 2016a*), even within the relatively small nervous system of *C. elegans,* composed of 118 neuron classes.

In this paper, we set out to test how broadly applicable the terminal selector concept is and to ask whether we can predict regulators of neuronal identity throughout the entire nervous system. To this end, we used two published datasets:

1. An expression atlas of the entire, fourth larval stage nervous system assembled through single cell transcriptomics (*Taylor et al., 2021*; *Figure 2A*, *Supplementary file 1B*) from which differential expression profiles were generated for each neuron class. In parallel, we used an available reporter gene atlas collectively generated by the *C. elegans* community over the past few decades via reporter gene analysis available at Wormbase.org (*Figure 2A*, *Supplementary file 1C*; *Hobert et al., 2016*).
2. The CISBP database of DNA recognition sites for 395 of 924 *C. elegans* TFs, 136 of 419 which are expressed in the differentiated nervous system and have been assigned unique DNA binding motifs (94 motifs were ambiguous) mainly through protein binding microarrays (PBMs) (*Narasimhan et al., 2015*; *Weirauch et al., 2014*; *Figure 2B*, *Figure 2—source data 1*; *Supplementary file 1A*) and supplemented by experimentally derived DNA binding motifs from our lab (*Etchberger et al., 2007*; *Kratsios et al., 2011*; *Masoudi et al., 2018*; *Wenick and Hobert, 2004*).

We used this neuron-specific gene expression atlas and TF DNA binding data in combination with a bioinformatic platform that we developed, TargetOrtho2, to examine whether gene batteries of specific neuron types display enrichment of DNA binding sites for putative neuronal identity regulators (*Figure 1A*). We indeed find such enrichment among the gene batteries of all individual neuron types. Therefore, our analysis (a) provides strong support for the concept of terminal selector TFs in the *C. elegans* nervous system and (b) predicts novel terminal selectors for many neuronally expressed TFs for most neuron types of the nervous system. The prevalence of such regulatory logic in the *C. elegans* nervous system suggests that such regulatory logic may be widespread in other organisms as well.

# Results

## Prediction of TF target genes using supervised learning: the TargetOrtho2 pipeline

Due to their relatively small size, predicted TF binding sites can be found abundantly scattered throughout the genome. We reasoned that the likelihood of a TF binding site bearing relevance for the activation of a nearby gene is much increased if a site is phylogenetically conserved in the cross-species orthologs of that gene. We previously validated this assumption in the following manner: we generated a software pipeline, CisOrtho, that interrogated the genome of two nematode species, *C. elegans* and *Caenorhabditis briggsae*, for binding sites for the heteromeric TTX-3::CEH-10 complex that is exclusively expressed in the AIY interneurons (called 'AIY motif') (*Bigelow et al., 2004*). Orthologous genes in *C. elegans* and *C. briggsae* that contain conserved copies of the AIY motif were indeed confirmed to be expressed in AIY (*Bigelow et al., 2004*; *Wenick and Hobert, 2004*). We further developed this phylogenetic footprinting approach, predicting and experimentally validating additional TF targets based on binding site conservation criteria (*Glenwinkel et al., 2014*). Based on these previous findings, we developed a novel software pipeline, called TargetOrtho2.

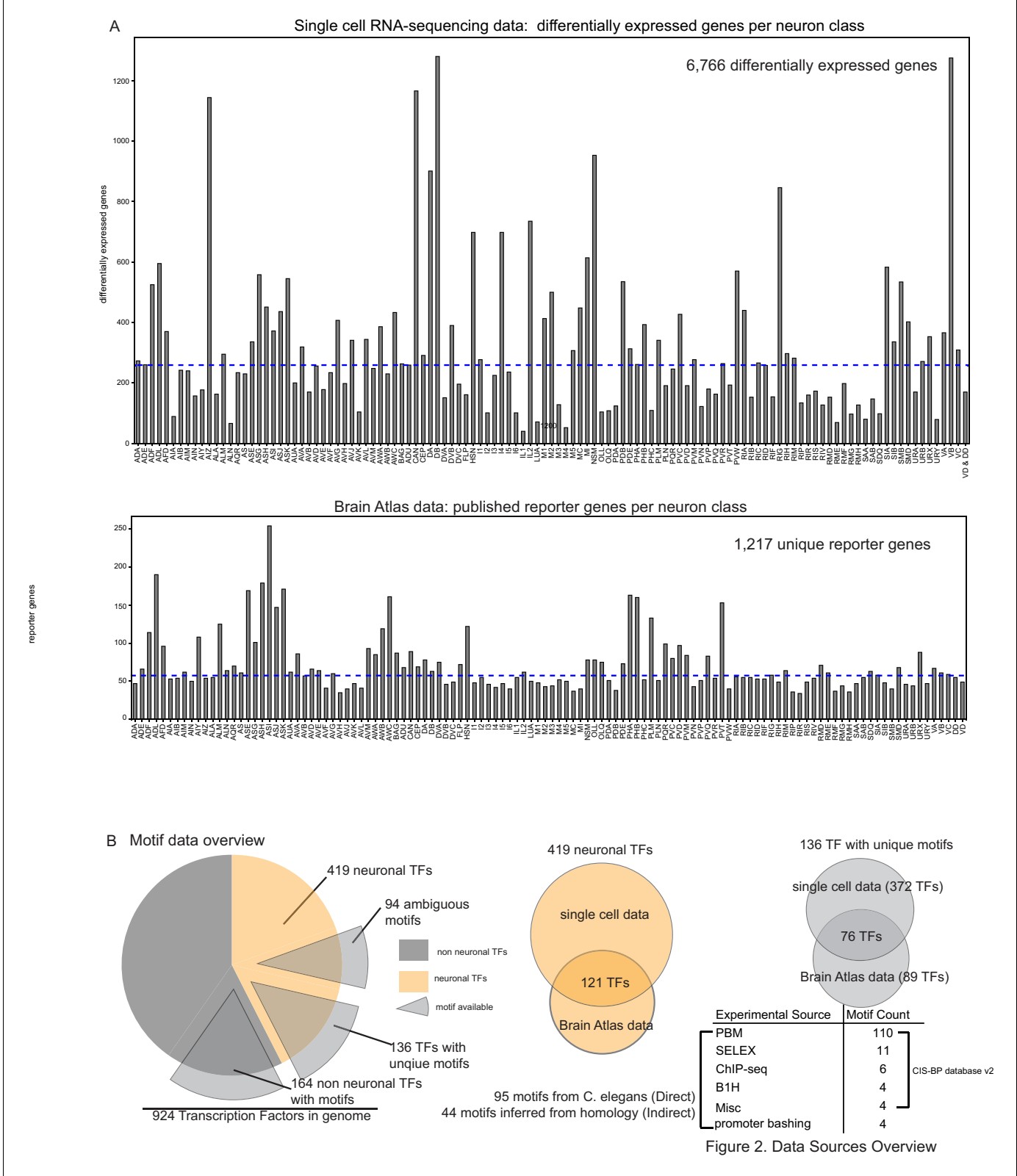

**Figure 2.** Data sources overview. (A) Top: Differentially expressed genes per single cell profile (*Taylor et al., 2021*). Bottom: Reporter count per neuron class from the Brain Atlas collection (a compendium of reporter expression patterns extracted from wormbasse.org; *Hobert et al., 2016*). (B) Motif data overview. Of 230 neuronal transcription factors (TFs) from the Brain Atlas collection and single cell differential expression profiles, 136 TF have unique

*Figure 2 continued on next page*

*Figure 2 continued*

motifs, 94 TFs have somewhat ambiguous motifs where more than one *Caenorhabditis elegans* TF has a similar DNA binding domain as computed in *Lambert et al., 2019*. The majority of motifs are derived from protein binding microarrays (PBMs) (see *Weirauch et al., 2014*).

The online version of this article includes the following source data for figure 2:

**Source data 1.** DNA binding motif logos.

TargetOrtho2 uses multiple species phylogeny and supervised learning to identify TF target genes from instances of binding sequences that are likely to be functionally relevant for gene expression.

As described in detail in the Materials and methods section, TargetOrtho2 is a tool for predicting TF target genes from putative TF binding sites in non-coding genomic sequences. Taking a position-specific scoring matrix (PSSM) as input, TargetOrtho2 uses FIMO (*Grant et al., 2011*) to find individual motif occurrences among the genomes of multiple species. Motif matches in upstream and intronic genome regions of orthologous genes from multiple nematode genomes are used to score candidate TF target genes by accounting for motif conservation in orthologs, frequency, and log-odds scores derived from the PSSM (*Figure 3A*). Candidate TF target genes are rank ordered based on motif feature data per gene. Rather than ranking candidate TF regulatory target genes based on non-weighted normalized motif feature scores as in the previous published version of TargetOrtho (*Glenwinkel et al., 2014*), we tested and implemented a supervised learning approach in which previous in vivo validated TF target gene (*Figure 1B*) motif features are used to train a classifier for predicting and ranking novel TF target genes (*Figure 3B*). To evaluate which supervised learning approach was superior in distinguishing true TF target genes from random genes in the genome with a DNA binding motif present, we employed a cross-validation scheme in which motif features from validated TF target genes (genes shown in *Figure 1B*) were shuffled together with random genes with a DNA binding motif. These shuffled genes were then split into a training set and validation set (*Figure 3—figure supplement 1*).

We evaluated 14 classifiers for the ability to distinguish true TF target genes from random genes with motif matches. While several classifiers performed well including the linear regression and random forest classifiers, we found that a Gaussian process classifier (GPC) trained on motif features from UNC-3, CHE-1, and TTX-3::CEH-10 validated target gene motif feature data best distinguished true TF target genes from random genes (*Figure 3—figure supplement 1*). In general, GPCs tend to perform well even when the underlying data does not meet the underlying assumption of normality making them suitable for analyzing motif feature data. TargetOrtho2 (*Figure 3C*) employs the GPC trained on this same motif feature data (*Figure 3B*) to assign probabilities and rank order candidate TF target genes for novel TFs with available DNA binding motifs. The final TargetOrtho2 output rank orders thousands of candidate TF target genes across the genome based on classifier probability outputs per gene (probability that gene is a 'true' target versus a 'random' gene) (*Figure 3D*). A detailed description of the supervised learning approach including the cross-validation procedure, motif feature selection process, and model evaluation are detailed in the Materials and methods section.

Recently, updated annotations have become available for several nematode genomes. TargetOrtho2 scans eight nematode genomes including members of diverse clades separated by 500 million years of evolution (*Caenorhabditis elegans, Caenorhabditis briggsae, Caenorhabditis remanei, Caenorhabditis japonica, Pristionchus pacificus, Pristionchus exspectatus, Ascaris lumbricoides*) (*Figure 3A*). The updated TargetOrtho2 software runs on OS X or Linux operating systems (*Figure 3C*). See Materials and methods for more TargetOrtho2 details including availability.

## TFs with previously described effects on select identity features are predicted to broadly control neuron type-specific gene batteries

Having established a pipeline to examine neuron type-specific gene batteries for the enrichment of phylogenetically conserved TF binding sites, using well-characterized TFs, we then considered 26 TFs for which (a) DNA recognition motifs are known, (b) some evidence was already available for their involvement in neuron type specification, and (c) that are continously expressed throughout the life of a neuron, a key feature of terminal selectors (*Supplementary file 2B*; *Figure 4A*). For example, the TF UNC-86 is required for the expression of 12 markers in the BDU neuron class

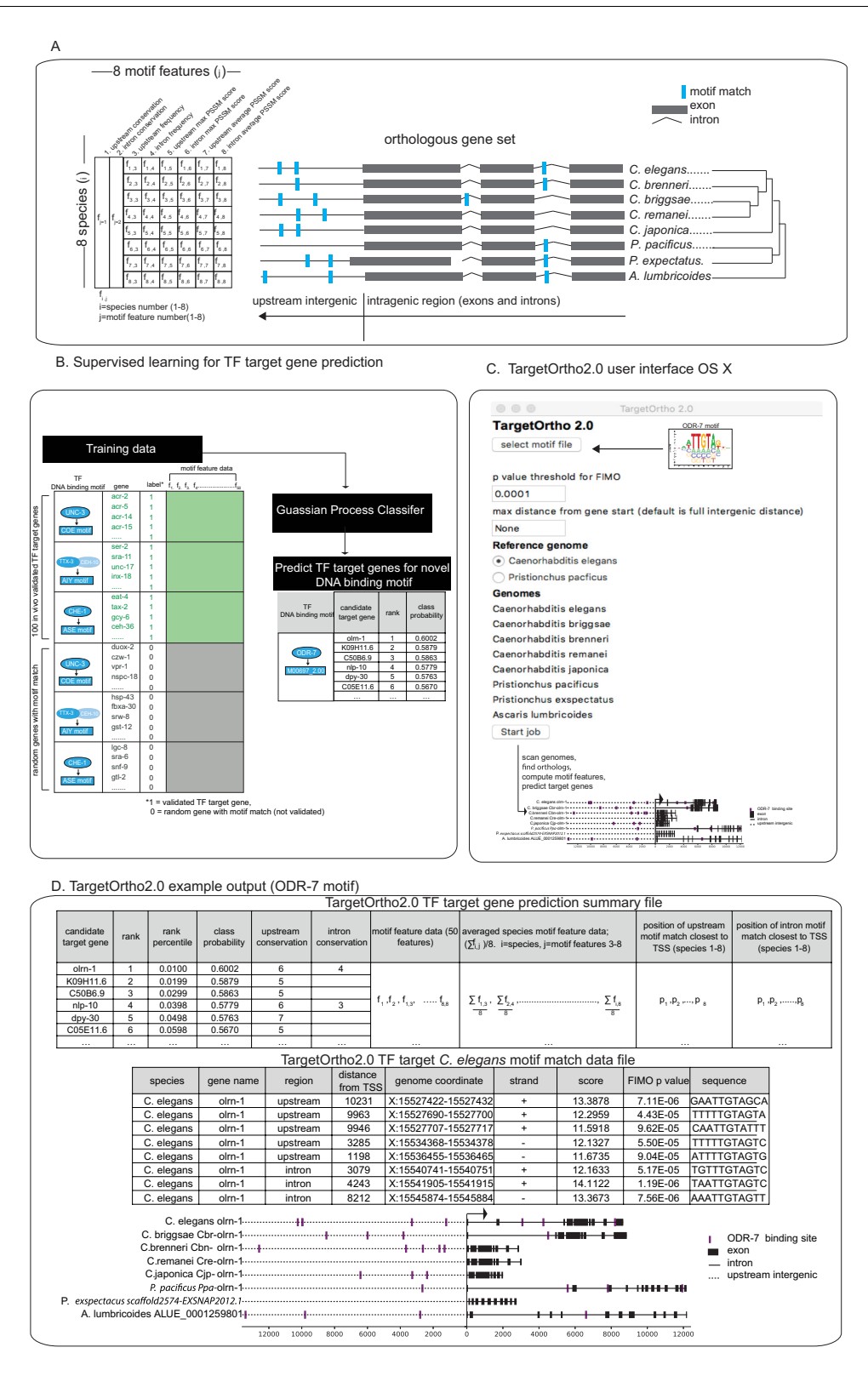

**Figure 3.** TargetOrtho2 development. (**A**) Ortholog schematic for eight nematode genomes included in TargetOrtho2. Fifty motif features used by TargetOrtho2's classifier to rank candidate transcription factor (TF) target genes are shown (left). TargetOrtho2 uses the FIMO scanner (**Grant et al., 2011**) to identify motif matches across eight nematode genomes. After assignment of each motif to the nearest gene loci, the following features are used to rank each potential TF target gene. $f_1$ and $f_2$ = upstream and intronic conservation where conservation is an integer corresponding to the

*Figure 3 continued*

number of species with at least one motif match (range 1 to 8). Other motif features used for classification are listed above. $f_{1,1}$=species 1, feature 1 corresponding to the species (label on right) and the motif feature (labeled above). Upstream frequency: motif match count upstream of gene transcription start site. Intron frequency: motif match count in all introns. Upstream max position-specific scoring matrix (PSSM) score: highest scoring upstream motif match (PSSM score from FIMO motif scanner; *Grant et al., 2011*). Intron max PSSM score: highest scoring intronic motif match. Upstream average PSSM score: average of all motif match PSSM scores upstream. Intron average PSSM score: average all motif match PSSM scores in introns. (B) Supervised learning for TF target gene prediction. Candidate TF target genes are rank ordered based on motif feature data per gene. Rather than ranking candidate TF regulatory target genes based on non-weighted normalized motif feature scores as in the previous published version of TargetOrtho2 (*Glenwinkel et al., 2014*), we tested and implemented a supervised learning approach in which previous in vivo validated TF target gene' (*Figure 1B*) motif features are used to train a classifier for predicting and ranking novel TF target genes. (C) TargetOrtho2 user interface for OS X. The user uploads a MEME formatted DNA binding motif file (*Bailey et al., 2009*) for a TF of interest and selects a p-value threshold for the FIMO motif scanner. The reference genome can be set to *Caenorhabditis elegans* or *Pristionchus pacificus* so that target genes statistics are output for a specific species. The search distance can be restricted to a user-defined upstream distance. If the intergenic distance is smaller than the selected distance, the smaller distance will be searched. TargetOrtho2 is also available as a command line tool for Linux with additional adjustable parameters. (D) Snapshot of TargetOrtho2 output from the ODR-7 motif. TargetOrtho2 outputs a summary file listing each candidate target gene and its relative rank order derived from the Gaussian process classifier (GPC) classifier. Each motif feature listed in part A is output in this file. An additional file showing each motif match per candidate target gene is output showing the PSSM scores and p-values from the FIMO scanner (*Grant et al., 2011*) as well as the matching DNA sequence and genome coordinate. Motif match information is output separately for each of the eight species.

The online version of this article includes the following figure supplement(s) for figure 3:

**Figure supplement 1.** TargetOrtho2 development cross-validation results.

**Figure supplement 2.** Four *bona fide* terminal selectors are enriched for phylogenetically conserved transcription factor (TF) binding sites (Brain Atlas data).

**Figure supplement 3.** Comparison of in vivo and in vitro derived transcription factor (TF) DNA binding motifs.

(*Gordon and Hobert, 2015*). We examined whether the 259 differentially expressed genes in this neuron class (log-fold change >0, p<0.05), based on single cell RNA sequencing (scRNA) analysis (*Taylor et al., 2021*), display an enrichment of phylogenetically conserved binding sites that are also significantly ranked by TargetOrtho2 as predicted target genes. We find 90% of differentially expressed BDU genes have UNC-86 motif matches (motif M03312_2.00) showing significant enrichment compared to genome-wide motif occurrences per gene (hypergeometric test p=1.54e-12).

Extending this analysis to the 26 TFs with known DNA recognition motifs and with genetic evidence of neuron differentiation roles (i.e. candidate terminal selectors), we find that neuron class-specific gene batteries are enriched and significantly rank ordered by TargetOrtho2 compared to random genes with a DNA binding motif in the genome for 24 out of 26 TFs in 60 out of 67 neuron class gene batteries examined (*Figure 4A* blue, *Supplementary file 2A,B*) (note that not all 75 neuron classes with prior evidence of a terminal selector [*Figure 1D*] had a determined DNA binding motif for that terminal selector). For example, binding sites for the putative terminal selector of dopaminergic neurons, the Dll/DLX ortholog CEH-43, which we had previously shown to regulate a few markers of all *C. elegans* dopaminergic neurons (*Doitsidou et al., 2013*), are indeed enriched in the dopaminergic effector gene battery (*Figure 4B*. *Supplementary file 2A*). Similarly, binding sites for the putative terminal selector NHR-67, a Tll/TLX ortholog, which we had previously shown to regulate markers of a subset of *C. elegans* GABAergic neurons (AVL, RIS) (*Gendrel et al., 2016*), are indeed enriched in the terminal gene batteries of these two neurons (*Figure 4B*. TableS2A).

In total, this analysis revealed three distinct motif signatures among neuron class-specific gene batteries (diagramed in *Figure 4B*):

1. A 'coordinated regulatory' signature (*Figure 4B*, blue) in which resident genes are both significantly enriched for DNA binding motifs compared to genes across the entire genome (*Figure 4B*, enrichment test p<0.05) and also significantly rank ordered (Wilcoxon rank sums test p<0.05) compared to random genes across the genome with a DNA binding motif (*Figure 4B*, gene rank test). This signature is consistent with a terminal selector-based regulatory model in which a large cohort of terminal identity genes are coordinately regulated by the same TF (as schematized in Fig.1B).

2. A 'piecemeal regulatory' signature in which resident genes are not significantly enriched, but those genes that do have a DNA binding motif match are significantly rank ordered by TargetOrtho2 compared to random genes across the genome with a motif (*Figure 4B* green).

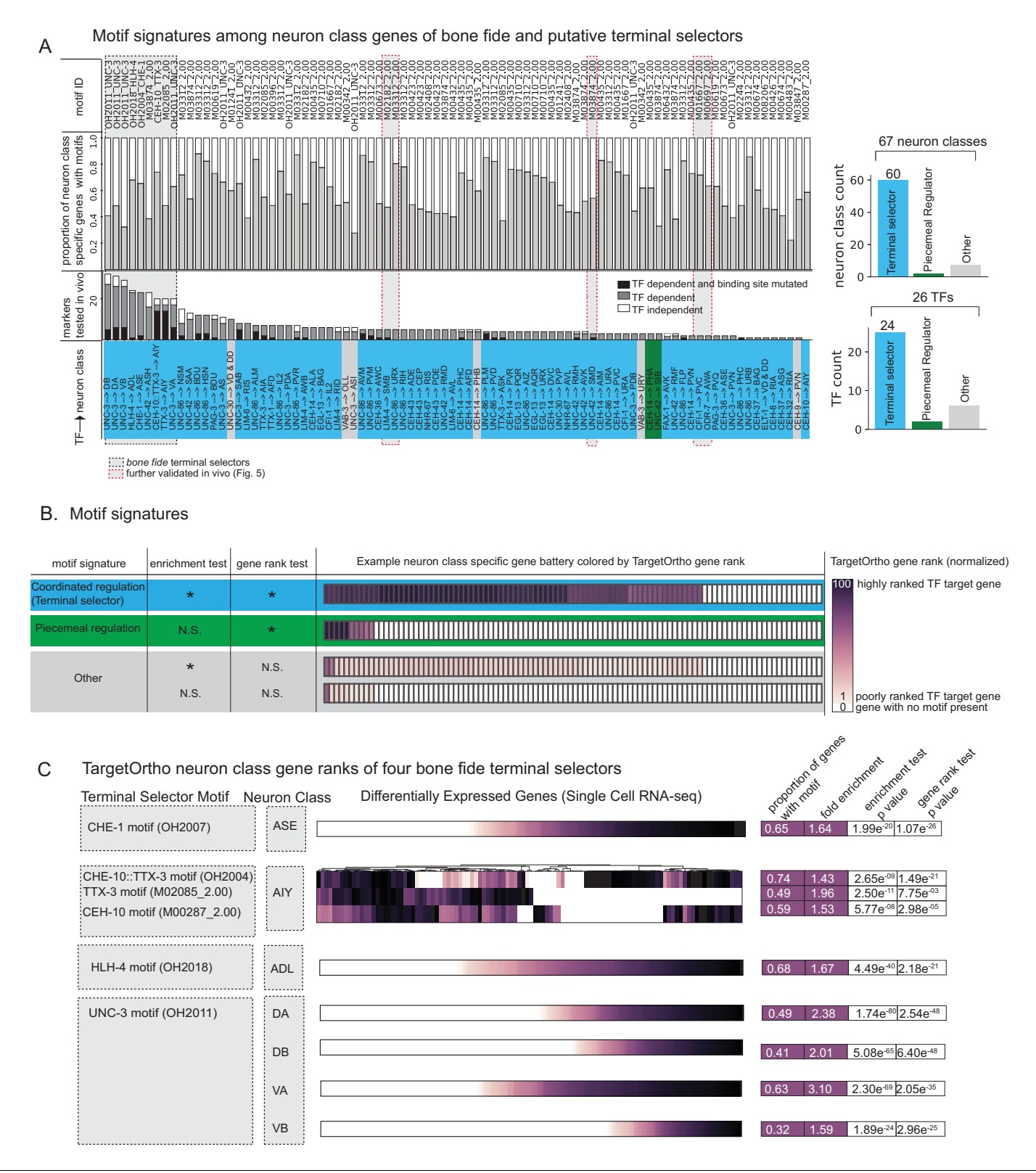

**Figure 4.** Terminal selectors with published genetic loss-of-function data. (**A**) Transcription factors (TFs) with previously described effects on select identity features are predicted to broadly control neuron type-specific gene batteries. TFs and target neuron class-specific gene batteries are listed with the proportion of markers tested in the corresponding genetic loss-of-function mutant. TFs and neuron class labels are colored according to motif signatures described in 4B. Only putative terminal selector TFs with DNA binding motifs and prior evidence of direct regulation of neuron class-specific

*Figure 4 continued on next page*

*Figure 4 continued*

genes are shown regardless of motif enrichment status. (B) DNA binding motif signatures in neuron class-specific gene batteries. Three distinct motif signatures among neuron class-specific gene batteries are diagrammed: (1) a 'coordinated regulatory' motif signature (blue) in which resident genes are both significantly enriched for DNA binding motifs compared to genes across the entire genome (enrichment test p<0.05) and also significantly rank ordered (Wilcoxon rank sums test p<0.05) compared to random genes across the genome with a DNA binding motif (gene rank test); (2) a 'piecemeal regulatory' motif signature in which resident genes are not significantly enriched, but those genes that do have a DNA binding motif match are significantly rank ordered by TargetOrtho2 compared to random genes across the genome with a motif (green); and (3) 'other' (gray), a situation where many resident genes have a motif match, but these genes are not significantly rank ordered by TargetOrtho2 as likely TF target genes or where residents are neither enriched for motifs nor rank ordered by TargetOrtho2. The corresponding tests are indicated in the table to the left. *p<0.05, N.S. = not significant. Enrichment test = hypergeometric test for enrichment. Rank sums test = Wilcoxon rank sums test. Example neuron class-specific gene batteries are colored by TargetOrtho2 rank order per gene where highly ranked TF target genes are dark purple and poorly ranked TF target genes are lighter purple. White boxes indicate genes with no DNA binding motif match present. TargetOrtho2 rankings are normalized from full genome DNA binding motif matches where the best predicted TF targets are assigned a value of 100 and the worst are assigned a value of 1. (C) Four *bona fide* terminal selectors are enriched for phylogenetically conserved TF binding sites. DNA binding motif matches in differentially expressed neuron class genes colored by normalized TargetOrtho2 rank order (see color bar in part B). UNC-3 motif (OH2011) from *Kratsios et al., 2011*, ASE motif (OH2007) from *Etchberger et al., 2007*, AIY motif (OH2004) from *Wenick and Hobert, 2004*, ADL motif (OH2018) from *Masoudi et al., 2018*. Darker shades of purple are better ranked by TargetOrtho2 as TF target genes. Lighter purple genes have weak motif matches and white have no motif matches. Three motifs are available for the AIY. The in vivo derived AIY motif (OH2004) and the PBM derived TTX-3 and CEH-10 motifs. AIY genes are clustered for visualization of motif match overlap. Right. The proportion of markers with a motif, fold enrichment from hypergeometric test for enrichment and p-values from the hypergeometric and Wilcoxon rank sums tests for motif enrichment and TargetOrtho2 ranking, respectively.

The online version of this article includes the following figure supplement(s) for figure 4:

**Figure supplement 1.** Proportion of predicted coordinated regulators versus number of markers tested.
**Figure supplement 2.** Comparison of in vivo and in vitro derived TF DNA binding motifs.
**Figure supplement 3.** Four bona fide terminal selectors are enriched for phylogenetically conserved TF binding sites (Brain Atlas data).
**Figure supplement 4.** Motif presence in published, TF-dependent reporter genes.

3. 'Other' (*Figure 4B*, gray), a situation where many genes have a motif match, but these genes are not significantly rank ordered by TargetOrtho2 as likely TF target genes or are neither enriched for motifs nor rank ordered by TargetOrtho2.

Four well-characterized terminal selectors, TTX-3::CEH-10, HLH-4, and UNC-3, have confirmed coordinated regulatory motif signatures in the ASE, AIY, ADL, and cholinergic motor neuron classes (DA/DB/VA/VB) (*Figure 4C*). That these particular neuron class-specific gene batteries determined from single cell expression experiments are enriched for highly ranked DNA binding motifs was not a foregone conclusion and further corroborates that terminal selectors likely directly bind their regulatory targets to confer cell type identities.

We set out to experimentally probe the putative terminal selector activity of three TFs that had coordinated regulatory motif signatures and five or fewer reporters examined in loss-of-function mutants. We call these TFs 'putative' terminal selectors due to the paucity of supporting genetic evidence in the literature. We examined the expression of a number of predicted TF target genes from TargetOrtho2 in specific mutant backgrounds: ODR-7 in AWA, UNC-86 in URX, and CFI-1 in PVC.

The *odr-7* orphan nuclear hormone receptor was previously shown to regulate expression of the *odr-10* gene in the AWA neurons (*Sengupta et al., 1996*; *Sengupta et al., 1994*) and to autoregulate as well (*Colosimo et al., 2003*). TargetOrtho2 identifies phylogenetically conserved ODR-7 binding sites in both *odr-10* and *odr-7* itself and predicts significant enrichment (hypergeometric p=1.04e-7) and ranking (Wilcoxon p=2.04e-8) of gene loci among the other 353 AWA differentially expressed genes (as determined by scRNA sequencing; *Taylor et al., 2021*). ODR-7 is among the top two terminal selector candidates for AWA identity (*Supplementary file 2A*, *Supplementary file 2F*). We generated reporters for two predicted ODR-7 target genes, *ins-1* and *pgp-2*, and found that both express in AWA, and that their expression is abrogated in *odr-7* mutant animals (*Figure 5*, *Supplementary file 2C*). This corroborates the case for ODR-7 being a terminal selector of AWA neuron identity.

Similarly, the ARID-type TF *cfi-1* was previously found to regulate the expression of two ionotropic glutamate receptors in the PVC interneurons, *nmr-1* and *glr-1* (*Shaham and Bargmann, 2002*). TargetOrtho2 identifies phylogenetically conserved CFI-1 binding sites in both genes and we find significant enrichment and TargetOrtho2 ranking of PVC differentially expressed genes overall (*Figure 5*, *Supplementary file 2A*). We asked whether the three genes that define the cholinergic

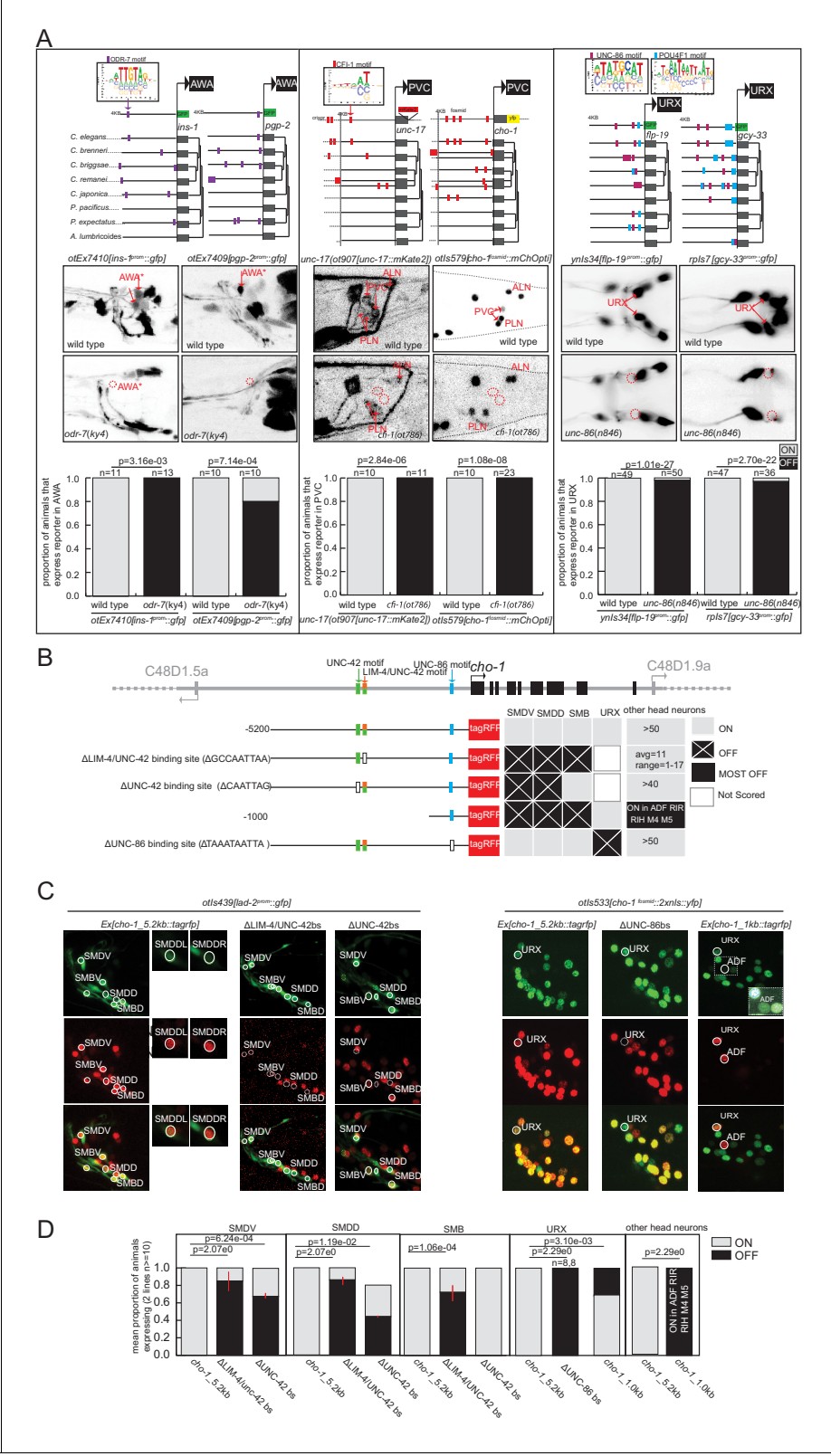

**Figure 5.** Experimental validation of putative terminal selectors. (**A**) Genetic loss-of-function mutant analysis for three transcription factors (TFs) with little previous evidence of direct regulation. ODR-7 is required for expression of *ins-1* and *pgp-2* reporters in AWA. CFI-1 is required for expression of *unc-17* and *cho-1* in PVC, and UNC-86 is required for expression of *flp-19* and *gcy-33* in URX. (**B–D**) Validation of *cis*-regulatory motifs. (**B**) Schematic of UNC-42 (green rectangle), LIM-4 (orange rectangle), and UNC −86 (blue rectangle) binding sites in the *cho-1* locus. White rectangles: binding site

*Figure 5 continued on next page*

Figure 5 continued

deleted (UNC-42 and LIM-4), or mutated (UNC-86). Black lines: upstream intergenic region. (C) *cho-1* reporter expression in SMD and SMB is lost when LIM-4/UNC-42 binding site is deleted. Deletion of the more proximal UNC-42 binding site results in loss of cho-1 reporter expression in SMD only. A *lad-2* reporter was used to identify SMD and SMB neuron classes. Mutation of the UNC-86 binding site results in loss of expression in URX. A 1 KB fragment is sufficient to drive cho-1 expression in URX and ADF. A *cho-1* fosmid reporter was used to identify URX. White dashed rectangle: enlargement and brightness increased to show dim gfp expression of *cho-1* in ADF. (D) Quantification of reporter expression in adults.

phenotype of PVC require *cfi-1* for expression in PVC. These include the transporter and enzyme *unc-17/VAChT* and *cha-1/ChaT* (which are organized into an operon-like structure) and the choline transporter *cho-1*. Each of these loci contains phylogenetically conserved, TargetOrtho2-predicted bindings sites for CFI-1 and we observe loss of expression of reporters for these genes in the PVC neurons of *cfi-1* mutants (*Figure 5*, *Supplementary file 2A*). This supports a role for CFI-1 as a terminal selector in PVC.

As a third example, we considered *unc-86* POU homeobox gene function in the URX neuron class. Previous work had shown that *unc-86* controls cholinergic identity features of URX (*Pereira et al., 2015*), as well as the expression of a soluble oxygen sensor, *gcy-32* (*Qin and Powell-Coffman, 2004*), and the neuropeptide *flp-8* (*Kim and Li, 2004*). Two binding sites are available for UNC-86, one derived from in vitro PBM data from the *C. elegans* protein, the other inferred from the human POU4F1 protein PBM data (*Supplementary file 1A*). We chose to include both motifs based on evidence from the literature that suggests the human protein derived motif closely matches the UNC-86 binding site that has been mutated in vivo and is required for expression of *unc-86*-dependent markers of touch receptor neuron identity (*Duggan et al., 1998*, *Leyva-Díaz et al., 2020*). We find that the POU4F1 derived binding motif (M03312_2.00) appears in over 80% of URX genes while the alternative *C. elegans* derived motif (M02244_2.00) appears in 44% of URX genes. Using TargetOrtho2 we find that phylogenetically conserved copies of both motifs are enriched in the scRNA-determined URX-specific gene battery (*Supplementary file 2B*). Apart from the above-mentioned genes, we examined whether the expression of two additional predicted UNC-86 target genes identified by TargetOrtho2, *flp-19* and *gcy-33*, also require UNC-86 for expression in URX. By crossing the corresponding reporter transgenes into an *unc-86* mutant background, we indeed found this to be the case (*Figure 5*).

## Further in vivo evidence of binding site usage by terminal selector TFs

As a complementary approach to the genetic loss-of-function analysis described above, we also tested the in vivo requirement of predicted TF binding sites of several putative terminal selectors with coordinated regulatory motif signatures in specific neuron classes. Specifically, we mutated the predicted UNC-86 binding site (motif M03312_2.00) in a transgenic reporter of the choline transporter gene *cho-1* and found complete loss of expression in URX (*Figure 5B–D*) suggesting this binding site is functional in URX. *cho-1* contributes to the neurotransmitter identity of many neuron classes for which we predict candidate terminal selectors, we therefore further examined the upstream sequence for other candidate terminal selector binding sites. We observed that *cho-1* is also a predicted target gene of LIM-4 and UNC-42, putative terminal selectors based on genetic loss-of-function data for the SMB (LIM-4: *Kim et al., 2015*) and SMD (UNC-42: *Berghoff et al., 2021*) neuron classes. Analysis of TargetOrtho2 data reveals coordinated regulatory motif signatures for each corroborating they are terminal selectors in the SMD and SMB neuron classes (*Supplementary file 2A*).

To assess the functional relevance of these predictions, we introduced a deletion of the LIM-4 binding site in the *cho-1* reporter construct and found this mutation caused loss of expression in SMB neurons, as predicted by genetic loss-of-function analysis (*Kim et al., 2015*). We also observed loss of *cho-1* expression in the SMD neuron class, supporting the bioinformatic prediction that LIM-4 may also be a terminal selector in SMD. However, loss of expression in SMD may also be attributable, at least in part, to the abrogation of the weak UNC-42 binding site that coincides with the LIM-4 binding site. Deletion of a second higher scoring UNC-42 binding site in the *cho-1* promoter results in loss of expression only in SMD neurons (SMB is not affected) (*Figure 5B–D*), as predicted by genetic loss-of-function analysis (*Pereira et al., 2015*). The requirement of these binding sites for expression of this cholinergic neuronal identity gene supports our predicted terminal selector role

for *unc-86*, *lim-4*, and *unc-42* in URX, SMB, and SMD neuron classes, respectively, and also demonstrates that in vitro derived DNA binding motifs used in this study predict biologically relevant TF binding sites.

Additional functional validation of our terminal selector prediction is provided in *Berghoff et al., 2021*, where we show that the UNC-42 homeodomain TF indeed controls the identity of the many neurons in which it is expressed, with the gene batteries of most of these neurons showing a significant DNA binding motif enrichment and TargetOrtho2 rank order of UNC-42 binding sites (*Berghoff et al., 2021* accompanying manuscript).

## Prediction of new regulators of neuronal identity throughout the entire *C. elegans* nervous system

To identify new regulators of neuronal identity, we examined DNA binding motif signatures (diagramed in *Figure 4B*) among scRNA-seq derived differential neuron class gene expression sets for all neuron types (*Figure 2A*). We gave no preference for whether a TF has previously been found to be involved in neuronal identity specification. Our only criteria for analyzing TFs were that (a) binding sites were available for our enrichment analysis and (b) those TFs were expressed throughout the life of a neuron, a key feature of candidate terminal selectors. Such continuous expression is implicit in our choice of TFs, because we inferred expression of a TF from scRNA data that was collected from fourth larval stage animals (*Taylor et al., 2021*).

We find that the differentially expressed gene batteries of 117 of 118 neuron classes have putative terminal selector TFs, based on each displaying one or more 'coordinated regulatory' motif signatures (*Figure 6A*, *Supplementary file 2C*). Each neuron class-specific gene battery shows enrichment for multiple distinct DNA binding motifs (up to around 28; *Figure 6A*) with a total of 124 TFs contributing to these type 'coordinated regulatory' motif signatures (*Figure 6A*, *Supplementary file 2E,F*). Note that motif signatures were only examined in neuron class gene batteries that express the TF. Many of these TFs are homeodomain-type TFs which corroborate the recently proposed hypothesis that they are prominent regulators of neuronal identity throughout the entire nervous system (*Reilly et al., 2020*).

We note that even in the most extreme cases of neuron class gene batteries with coordinated regulatory motif signatures, not every gene of a neuron type-specific gene battery contains binding sites for a presumptive terminal selector (see *Supplementary file 2C* for motif match proportions per neuron class gene battery). This finding is expected based on past functional analysis of terminal selectors: between 8% and 36% of reporter genes whose expression is terminal selector-dependent based on mutant analysis have either no predicted binding site or a poorly ranked binding site (bottom 50% percentile of genome-wide candidate TF target genes) (*Figure 6—figure supplement 1*). For example, of 23 ASE-expressed reporter genes that lose expression in *che-1* mutant animals (*Etchberger et al., 2007*), four have no predicted CHE-1 binding site and four others are poorly ranked CHE-1 target genes by TargetOrtho2. Expression of such reporter genes is either initiated using sub-threshold binding sites (below the FIMO scanner threshold of p=1e-4) or by indirect regulation, in which a terminal selector regulates the expression of another TF that then controls the specific effector gene. Several examples of indirect, 'feedforward' regulation have been previously described (*Altun-Gultekin et al., 2001*; *Etchberger et al., 2007*; *Gordon and Hobert, 2015*; *Kratsios et al., 2011*). Thus, we argue that while terminal selectors tend to have an enrichment of binding sites in the neuron class gene batteries they regulate, implicating them in direct regulation, we also expect a degree of indirect regulation.

Set apart from the 'coordinated regulatory' motif signatures are 'piecemeal regulatory' motif signatures (*Figure 4B*, green). We observe that with increasing evidence of direct regulation (number of TF-dependent markers examined with reporters), the probability of observing a coordinated regulatory motif signature (putative terminal selector signature) increases. For example, the TF CEH-14 has just three reported target genes in PHA (*Serrano-Saiz et al., 2013*) and has a 'piecemeal regulatory' motif signature while TFs with more than 20 experimentally validated target genes all display a 'coordinated regulatory' motif signature (*Figure 4—figure supplement 1*). Several candidate terminal selectors with few markers examined in loss-of-function mutants display piecemeal regulatory motif signatures indicating these candidates may not be terminal selectors (*Figure 4A,B*, *Supplementary file 2A*). Other TFs with 'coordinated regulatory' motif signatures are better

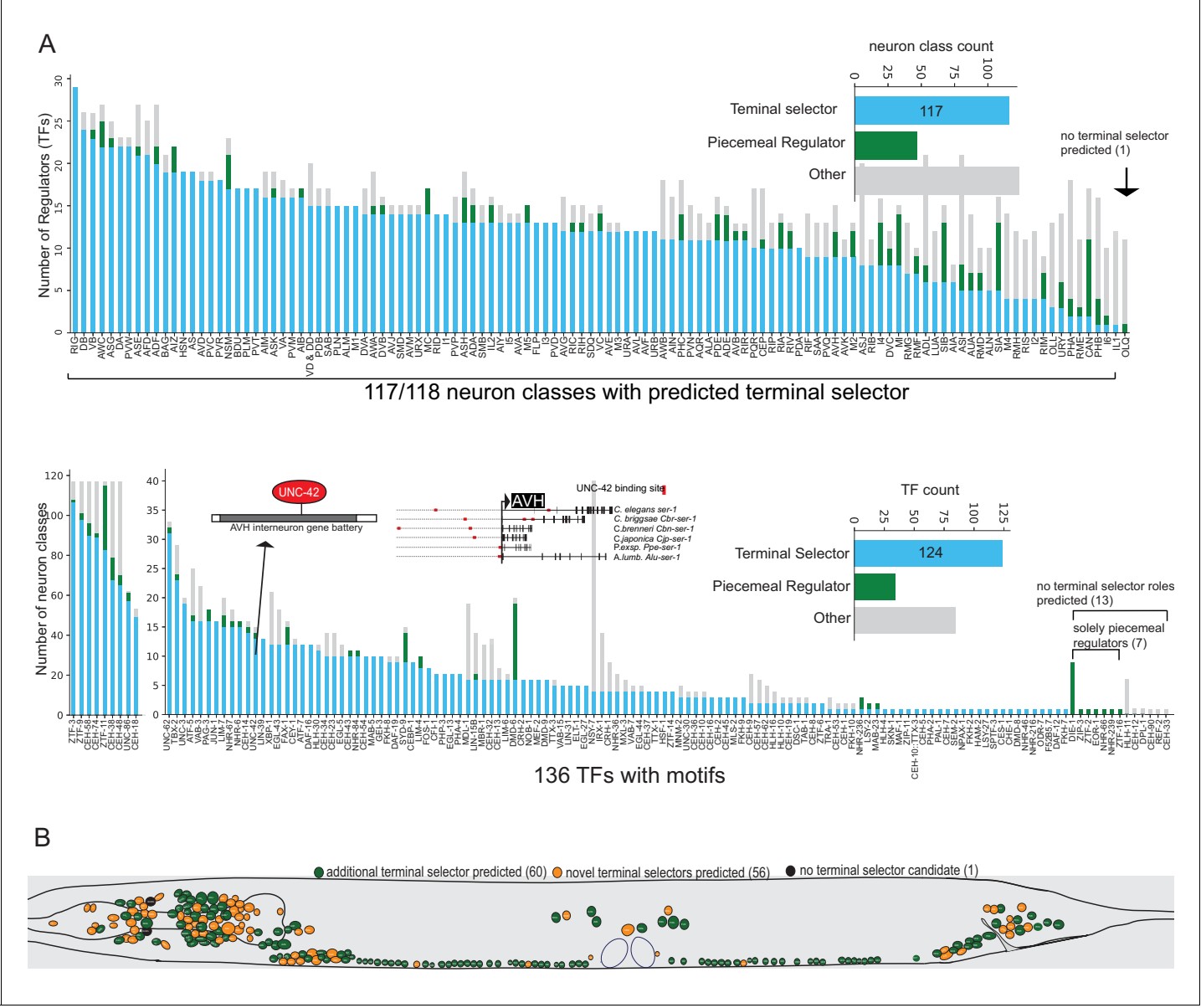

**Figure 6.** Motif signatures in the nervous system from single cell data. (**A**) Overview of all transcription factor (TF) motif signatures in differentially expressed neuron class genes across the nervous system. Most neuron classes have a candidate 'coordinated regulator' (see *Figure 4B*). Distribution of regulatory signatures by neuron class (top) and by TF (bottom). (**B**) Overview of predicted terminal selectors in the nervous system from single cell differential gene expression data. Terminal selectors refer to candidate coordinated regulators summarized in 6A.

The online version of this article includes the following figure supplement(s) for figure 6:

**Figure supplement 1.** Motif presence in published, transcription factor (TF)-dependent reporter genes.

candidate terminal selectors than the TFs with previous genetic loss-of-function data in these cases (see *Supplementary file 2F* for best candidate terminal selector per neuron class).

In most neuron classes, we observe a combination of 'coordinated regulatory' and 'piecemeal regulatory' motif signatures suggesting that neuron class gene batteries are regulated by both broad and targeted regulatory factors (*Figure 6A,B Supplementary file 2C,E,F*). The co-existence of such signatures may indicate the existence of hierarchical regulation in which a terminal selector coordinates the expression of most genes, including perhaps also piecemeal regulatory factors, to then jointly control target genes. Experimental evidence for such 'feedforward loops' exists in several neuron types (*Gordon and Hobert, 2015*; *Hobert, 2006*; *Hobert, 2016a*).

## Combinations of terminal selectors target common effector genes

As mentioned above, most neuron type-specific gene batteries display enrichments of more than one phylogenetically conserved TF binding site. In principle, one could therefore envision several distinct scenarios, illustrated in *Figure 7A*.

1. Co-occurrence of 'coordinated regulatory motifs' in common targets (*Figure 7A*, orange), such that observed DNA binding motif matches from each of two candidate terminal selectors are significantly co-enriched in common neuron class-specific genes compared to expected by chance (*Figure 7A*, co-enrichment test p<0.05). These common genes are each significantly rank ordered by TargetOrtho2 compared to random genes with DNA binding motif matches in the genome (*Figure 7A*, gene rank tests, p<0.05 for each motif).
2. Co-occurrence of 'piecemeal regulatory motifs' in common targets, in which significant co-enrichment and TargetOrtho2 rank order is observed as in the previous scenario, but one or both of the two TFs may be a piecemeal regulator (*Figure 7A*, yellow).
3. Lack of co-occurrence in common target genes, in which either significant co-enrichment and/or TargetOrtho2 rank order is not observed for one or both motifs (*Figure 7A*, gray).

To address these possibilities, we first focused on 17 cases where genetic loss-of-function analysis has already indicated the existence of multiple TFs that are each required to control the expression of common target genes in a specific neuron type (see *Hobert, 2016a* for references). This included, for example, the case of the heteromerically acting TTX-3 and CEH-10 homeodomain proteins (*Wenick and Hobert, 2004*). For 16 out of these 17 cases, we observed significant co-enrichment and ranking of TargetOrtho2-predicted target genes, suggesting that co-expressed terminal selectors with coordinated regulatory motif signatures mainly target common effector genes (*Figure 7B Supplementary file 2D*). For example, the CEH-14 LIM homeobox gene and the UNC-86 POU homeobox genes are predicted to co-regulate the gene battery of the PHC neurons based on the presence of co-enriched binding sites in the PHC gene battery and this prediction is validated by genetic analysis that showed loss of a number of PHC markers in *unc-86* and *ceh-14* mutants (*Serrano-Saiz et al., 2013*). Similarly, CEH-14 and CFI-1 binding sites are co-enriched in the same set of genes that constitute the effector gene battery of the cholinergic PVC neurons (see *Supplementary file 2D* for co-enrichment statistics) and genetic mutant analysis has previously confirmed loss of PVC marker gene expression in *ceh-14* mutants (*Pereira et al., 2015*), and, as we have shown above, *cfi-1* affects PVC marker genes (*Figure 6*). Taken together, we refer to this scenario as co-terminal selectors (*Figure 7B*).

We extended this analysis to all neuron classes throughout the nervous system. Rather than evaluating all possible combinations of TFs per neuron class for binding site co-enrichments entailing hundreds of thousands of possible combinations of cofactors and significance tests, we chose to evaluate cofactors in pairs, an approach that is computationally more feasible. One may examine the results of all co-enrichment significance tests to infer larger combinations of cofactor TFs (*Supplementary file 2G*). We examined pairs of candidate terminal selectors (defined in *Figure 4B*) expressed in the same neuron class and asked if these also tend to target common genes in a neuron type. Of these candidate terminal selector pairs (where motif 1 and motif 2 have a coordinated regulatory motif signature), we find that 98% show common target genes suggesting that a combinatorial activity of TFs is the predominant mode of effector gene regulation (*Figure 7C, D, E*, *Figure 7—source data 1*, *Supplementary file 2G*). For example, our analysis predicts that UNC-3 and UNC-42, rather than acting in parallel pathways, both target common effector genes in the command interneurons, a group of interconnected neurons controlling locomotion (*Figure 7C*, *Supplementary File S2G*). While many cofactor pairs are predicted to target common effector genes in one or several command interneurons, the UNC-3::UNC-42 pair uniquely shows co-motif enrichment in all command interneuron gene batteries. In an accompanying manuscript, we validate such cooperative function of *unc-42* and *unc-3* in specifying command interneuron identity (*Berghoff et al., 2021*). Another example is the predicted combinatorial activity of the CEH-8 and CEH-32 homeodomain TFs (*Figure 7D*, *Figure 7—source data 1*, *Supplementary file 2D,G*). Recent loss-of-function studies show that these two homeodomain TFs are required to specify the RIA interneurons (*Reilly et al., 2020*). We also catalog several instances in which co-expressed TFs are not predicted to co-regulate the same effector genes (*Figure 7D*). Broken down by neuron class, we observe predicted terminal selector co-regulators in which each individual TF is a predicted terminal selector

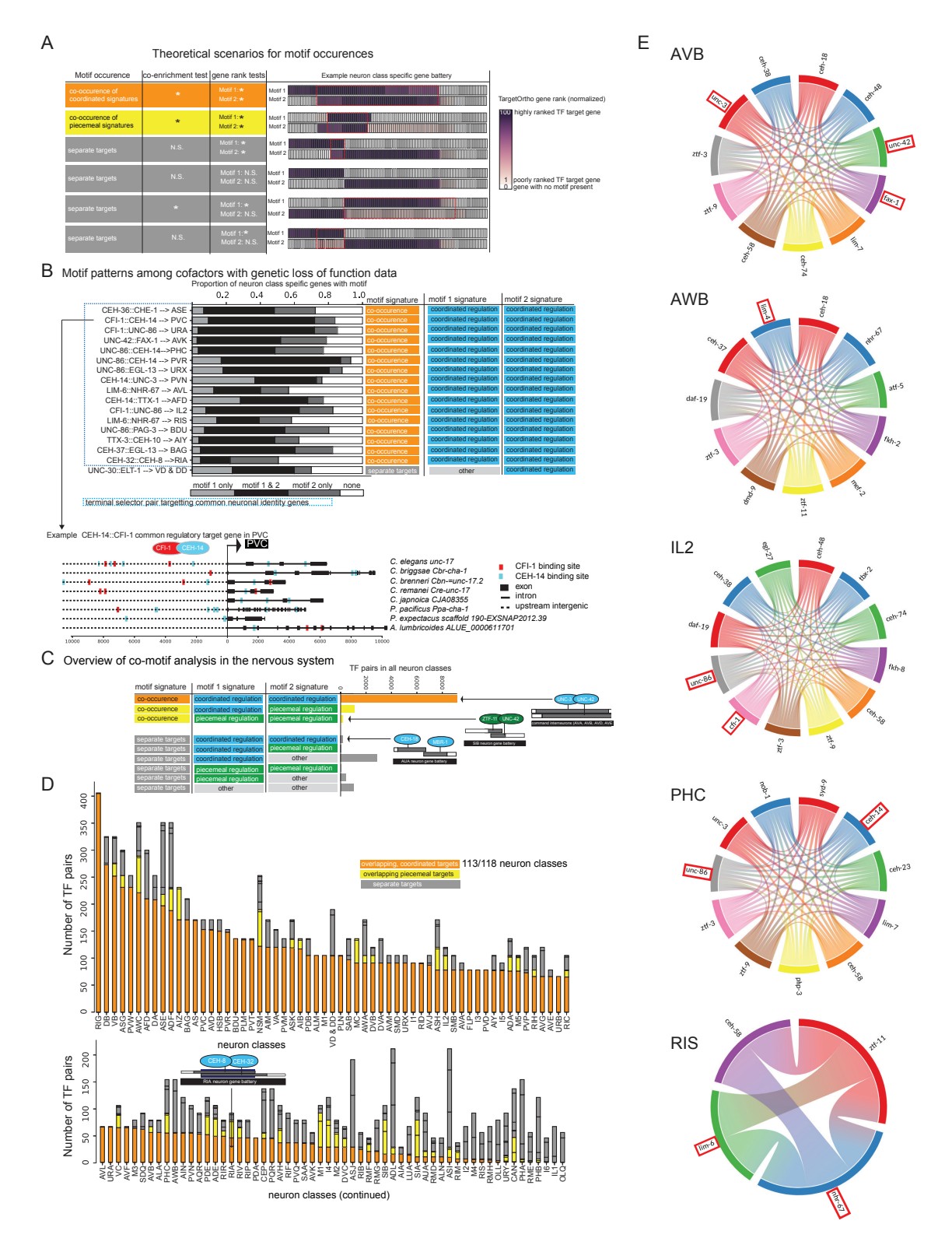

**Figure 7.** Terminal selector combinations inferred from motif co-occurences. (**A**) Co-motif signatures from pairwise cofactor analysis. (1) 'co-occuring coordinated signatures' (orange) in which not only is each TF individually characterized as a 'coordinated regulator' (*Figure 4A*, blue), but the observed DNA binding motif matches from each of two TFs are significantly co-enriched in common neuron class-specific genes compared to expected by chance (co-enrichment test p<0.05) and these common genes are each significantly rank ordered by TargetOrtho2 compared to random genes with

*Figure 7 continued on next page*

*Figure 7 continued*

DNA binding motif matches in the genome (gene rank tests, p<0.05 for each motif); (2) 'common regulators', in which significant co-enrichment and TargetOrtho2 rank order is observed as in case 1, but one or both of the two TFs may be a piecemeal regulatory (yellow); (3) 'independent regulators', in which either significant co-enrichment and/or TargetOrtho2 rank order is not observed for one or both motifs (gray). (B) Motif patterns among cofactors with genetic loss-of-function data. Colored labels on the right correspond to motif signatures of the individual TFs (right two-colored columns, see *Figure 4B*) as well as the co-motif signature (left colored column, see part 7A). Lower: Example of cofactor DNA binding motif matches in an orthologous gene set. (C) Cofactor analysis overview in the nervous system. Most cases where motif 1 and motif 2 have 'coordinated regulatory' motif signatures show common master regulator co-motif signatures. The master regulator TF pair targets common effector genes in a given neuron class gene battery (orange). Examples TF pairs shown to right. (D) Overview of cofactor motif signatures in the nervous system. Colors correspond to co-motif signatures described in part A. (E) Several examples of candidate terminal selectors whose binding sites are co-enriched in a given neuron class. The chord diagrams (generated using the Chord package for Python) indicate which combination of candidate terminal selectors show joined enrichment in neuron type-specific gene batteries. Chord diagrams of all other neurons in which co-enrichment of coordinated regulators were observed is shown in *Figure 7—source data 1*. TFs with genetic evidence for terminal selector function are boxed in red. All combinations are also listed in Supplementary File S2G. Note that TF expression is based on scRNA data and may contain false negative/positive signals (in contrast to transcripts, the UNC-86 protein is not expressed in ADL) and also includes broad/ubiquitously expressed TFs (e.g. ztf-3). This may inflate the number of putativte terminal selectors.

The online version of this article includes the following source data for figure 7:

**Source data 1.** Chord diagrams that shows predicted terminal selector combinations (with joined enrichment of binding sites in terminal gene batteries) for all neuron classes.

with common targets in 113 of 118 neuron classes (*Figure 7D, E*, *Figure 7—source data 1*, *Supplementary file 2D,G*).

In conclusion, our nervous system-wide motif analysis demonstrates that (a) most neuron classes have predicted coordinated regulators that may broadly co-regulate the many identify features that define individual neuron types and (b) coordinated regulatory cofactors expressed in the same neuron type tend to share common targets rather than regulate identity features in a piecemeal manner. Hence, our findings support the concept of coordinated control of terminal gene batteries by terminal selector complexes.

## Motif signatures in reporter constructs with validated neuron type-specific gene expression

As an orthogonal approach to the analysis of scRNA data, we analyzed a collection of approximately 1000 reporter genes, which we call 'Brain Atlas', manually curated from the literature at Wormbase (http://www.wormbase.org) and our own lab (*Hobert et al., 2016*). The expression patterns of this reporter collection cover all 118 neuron classes, with a median 57 of reporters expressed per neuron class (*Figure 2B*). These data show an incomplete overlap with the scRNA data, likely due to partial regulatory regions in reporters as well as sparse coverage of some neuron types by the scRNA datasets. We re-iterated our phylogenetic footprinting analysis to determine if the reporter dataset also reveals evidence of binding site enrichment and significant TargetOrtho2 rank ordering of neuron class gene batteries compared to random genes with DNA binding motif matches in the genome. We first checked the four *bona fide* terminal selector motif signatures among Brain Atlas reporters and find coordinated regulatory motif signatures among their seven target neuron classes (UNC-3 in DA,DB,VA,VB; CHE-1 in ASE; TTX-3::CEH-10 in AIY, HLH-4 in ADL) in agreement with motif analysis of corresponding single cell differential expression profile data (*Figure 3—figure supplement 2*, *Supplementary file 3A*). Our in vivo validations in *Figure 5* also show complete agreement of motif signatures among putative target neuron class gene batteries (*Supplementary file 3C*).

Examining all 118 neuron classes, we identified motif signatures for 136 neuronal TFs and predict terminal selectors (coordinated regulatory motif signatures) for all 118 neuron classes (*Supplementary file 3A*). A comparison between single cell and Brain Atlas datasets shows that the proportion of genes with motifs per neuron class is highly correlated (Spearman correlation = 0.80, p=0) (*Figure 8A*) despite the independent nature of each dataset (marker population overlap per neuron class varies from 0% to 11%, average 4.5%). Furthermore, the identified motif signature model per TF per neuron class is significantly correlated (Spearman correlation = 0.30, p=$2.66e^{-38}$) (*Figure 8B*). Examining the top two predicted terminal selectors per neuron class (sorted by rank sums test p-value), we find 83% agreement of specific terminal selectors between the datasets (excluding panneuronally expressed regulators) (*Figure 8C*, *Supplementary file 3B,C*). When

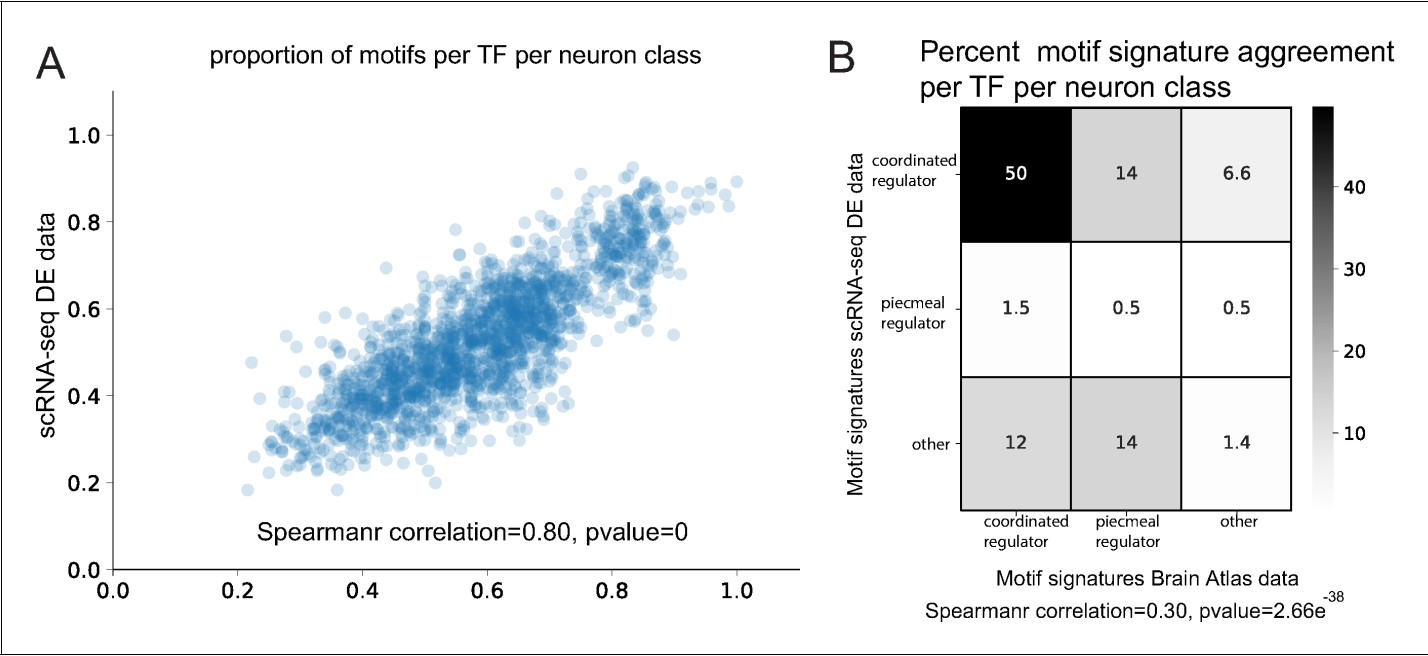

**Figure 8.** Comparison of Brain Atlas reporter-based motif analysis and single cell RNA-sequencing derived results. (**A**) The proportion of motifs per transcription factor (TF) per neuron class is significantly correlated between the two datasets. (**B**) Motif signature model assignments per TF per neuron class are significantly correlated between the two datasets.

panneuronally expressed regulators are included in this comparison, we find 92% agreement; this increase appears to be driven by *ztf-11*, a TF implicated neuronal transdifferentiation (*Lee et al., 2019*), predicted to have a coordinated regulatory role in 73 neuron classes. The remaining neuron classes predict common coordinated regulatory signature in the scRNA-seq and Brain Atlas datasets, but the order of top ranked regulators varies (*Supplementary file 3C*). The result of this parallel analysis of neuronal reporter genes corroborates a terminal selector role in the diversification of cell types in the *C. elegans* nervous system.

## Discussion

Several decades worth of genetic loss-of-function studies in *C. elegans* have uncovered a number of TFs required for neurons to acquire specific anatomical, functional, or molecular features (*Hobert, 2016a*). In most cases, the phenotypic analysis of such TF knockout strains was limited to a small number of molecular markers, hence leaving it unclear how broadly a TF affected the overall identity of a neuron. However, in some instances, these studies extended to a large array of functional and molecular identity features. In all of these cases, most if not all available molecular identity markers were found to be affected by specific TFs, leading to the classification of these TFs as 'master regulatory' TFs, or terminal selectors, that broadly define neuronal identity (*Hobert, 2008*). The key operational feature of terminal selectors is that they broadly affect terminally identity features of a neuron, without affecting the initial generation of the neuron (thereby setting them apart from earlier acting patterning genes) and that they are continuously expressed throughout the life of a neuron, thereby indicating roles in not only coordinating the initiation but also the maintenance of terminal identity features. Yet, even in the most deeply studied cases, the breadth by which these TFs affect, that is, coordinate, the expression of neuron type-specific gene batteries remained unclear. Do they indeed coordinate the expression of most of the terminal identity features of a neuron?

Here, we have described a systematic study of the potential range of the activity of individual TFs in individual neuron classes. Specifically, the recent availability of single cell expression profiles of the mature *C. elegans* nervous system (*Taylor et al., 2021*) and the elucidation of DNA binding

specificity of many *C. elegans* TFs (*Weirauch et al., 2014*) has enabled us to ask whether neuron type-specific gene batteries can indeed be expected to be controlled by a coordinated mechanism. It is important to recall that, a priori, different models could be envisioned for how neuron type-specific gene batteries are controlled ('piecemeal control' by many TFs versus 'coordinated regulation'; *Figure 1A*). Our analysis provides support for the coordinated, terminal selector-type model of neuron identity specification. The gene batteries of essentially all individual *C. elegans* neuron classes were found to display enrichment of phylogenetically conserved *cis*-regulatory motifs. Importantly, these motifs were not identified de novo, but were the in vitro-defined binding sites of TF(s) known to be expressed in the respective neuron class.

The purely computational identification of binding site enrichment alone would already be quite suggestive for a regulatory relationship between the TFs and the genes that contain these binding sites. Yet, a *de facto* experimental validation of such regulatory relationship has already been provided in the past through many genetic loss-of-function studies. For example, the gene battery expressed in the AWA olfactory neurons is enriched for the in vitro binding site of the nuclear hormone receptor ODR-7 and genetic loss-of-function studies had previously shown that ODR-7 regulates one such AWA-expressed effector gene with an ODR-7 binding site, the GPCR-encoding *odr-10* locus (*Sengupta et al., 1996*). We further validated here several additional loci as ODR-7 targets in vivo. Altogether, several dozen neuron classes were already associated with a TF that affected the expression of one or some very few targets (*Hobert, 2016a*). Based on our finding of enriched, phylogenetically conserved binding sites, we found that these TFs are also predicted to control many other specific features of the respective neuron class. Hence, a coordinated regulatory control mechanism appears to be the major mode by which neuronal identity specification is achieved in *C. elegans*.

Even though there is a significant enrichment of binding sites for terminal selectors in neuron type-specific gene batteries, not every gene battery member contains high-scoring, well-conserved binding sites. This could indicate either non-conventional binding or indirect regulation via intermediary factors. It is also possible that some members of a terminal gene battery are regulated instead by different factors, but if this were the case, such factors would only be responsible to regulate a subset of identity features. The enrichment of bindings sites – as well as extensive past genetic loss-of-function analysis – clearly underscores the predominant role of terminal selectors in neuronal identity control.

Another key conclusion from our study is that it provides and supports a concrete mechanistic basis for combinatorial control mechanisms in neuronal identity specification. Previous genetic loss-of-function studies have shown that in many different neuron types, the removal of not just one, but two or more TFs resulted in similar types of neuron identity specification defects. In the two best characterized cases, UNC-86 and MEC-3 homeodomain TFs bind cooperatively to adjacent binding sites in touch neuron-expressed genes (*Xue et al., 1993*) and the CEH-10 and TTX-3 homeodomain TFs bind cooperatively to adjacent binding sites in the AIY interneuron (*Wenick and Hobert, 2004*). The present study reveals that the terminal gene batteries of most neuron classes tend to show an enrichment of phylogenetically conserved sites for more than one predicted terminal selector, in several cases more than a dozen. Since the binding sites for such collaborating factors are co-enriched in the same target genes, these factors either bind DNA cooperatively and/or operate as a transcription factor collective on their target promoters . Many of these predicted cofactor combinations are homeodomain proteins, which is consistent with our recent finding that each individual neuron class in *C. elegans* expresses a unique combination of homeodomain TFs (*Reilly et al., 2020*). Our binding site analysis demonstrates that those 'homedomain codes' leave phylogenetically conserved, *cis*-regulatory footprints in neuron type-specific gene batteries and corroborate their critical role in neuronal identity specification.

A number of limitations of our studies are notable. First and foremost, our pipeline entirely relies on prior knowledge about the existence of TF binding sites; it is not designed for de novo motif discovery, a feature handled by other software suits including MEME (*Bailey et al., 2009*) or FIRE (*Elemento et al., 2007*). Not all *C. elegans* TFs currently have well-defined binding sites and, as such, our analysis likely underestimates the number of regulators of neuronal identity. Moreover, we excluded 94 DNA binding motifs (41% of 215 available motifs) that were highly similar to other *C. elegans* TFs to avoid ambiguous motif signatures where one or more TFs may use the same binding site. This circumstance is especially problematic if more than one such TF is expressed in the same

cell in which case it not possible to use DNA motif evidence to infer how broadly these TFs regulate their targets. Also, our (partial) reliance on in vitro derived DNA binding sites does not take into account the possibility that binding sites may differ in vivo. In spite of these limitations, our analysis provides strong support for the terminal selector model of neuron type specification and it provides a rich resource for future experimentation. In principle, prediction of neuronal regulators using this type of phylogenetic analysis can be undertaken in any cohort of well-conserved genomes with well-defined orthology assignments.

The breadth by which the coordinated regulatory principle appears to operate throughout the nervous system of *C. elegans* suggests that this principle may be used broadly in other organisms as well. The study of a number of TFs in the vertebrate nervous system suggests that coordinated regulation of terminal gene batteries by terminal selectors is indeed conserved across phylogeny (*Deneris and Hobert, 2014*), even though the evidence is currently limited. One general problem in revealing functions of TFs during terminal differentiation in more complex organisms is that many TFs are often utilized at multiple distinct steps during development, including early progenitor specification. Hence, simple, non-temporally controlled mutant analysis will obscure later functions. For example, vertebrate HOX cluster genes have functions in early patterning of the spinal cord, but their function and expression during later stages of terminal differentiation have, surprisingly, remained unstudied. In spite of these often encountered challenges, a few notable examples of putative vertebrate terminal selectors include the Ets domain TF Pet1, which in combination with other factors broadly coordinates expression of terminal identity markers of serotonergic neurons in the hindbrain (*Spencer and Deneris, 2017*), several Dlx homeodomain factors which directly control and maintain terminal identity features in forebrain GABAergic interneurons (*Lindtner et al., 2019*; *Pla et al., 2018*), the Lhx2 protein in olfactory neurons (*Monahan et al., 2017*; *Zhang et al., 2016*), Brn3a in the habenula (*Serrano-Saiz et al., 2018*), Dlx homeodomain proteins in forebrain GABAergic interneurons (*Lindtner et al., 2019*; *Pla et al., 2018*), the Pitx3 homeodomain protein in dopaminergic neurons (*Smidt and Burbach, 2009*), or the homeodomain protein Crx in in photoreceptors (*Corbo et al., 2010*).

## Materials and methods

### TargetOrtho2
#### Improvement in computation time
TargetOrtho2 is approximately 50 times faster than the previous version (*Glenwinkel et al., 2014*) with run times averaging 20 min per job (median 17.8 min, range 8 to 80 min, FIMO p-value threshold = 1e-4). This improvement in job processing time made this study possible with a reduction in processing time for the 100 plus neuronal TF motifs examined from 91 to 1.8 days.

#### Software changes in version 2.0
Increased processing time was achieved by switching from Python's sqlite3-based database programming to Pandas library. TargetOrtho2 uses the FIMO software (*Grant et al., 2011*) from MEME (*Bailey et al., 2009*) to scan eight species genomes. We implement the BEDOPS software (*Neph et al., 2012*) for assignment of motif match coordinates to adjacent annotated coding genes. BEDOPS allows a search of full intergenic and intronic regions when assigning a motif match to adjacent gene loci. Binding sites in downstream and exonic regions are also annotated and output with TargetOrtho2 results but these values are not used for ranking of candidate TF target genes.

#### Implementation of supervised learning
We improved TF target gene prediction from motif feature data by implementing a supervised learning algorithm in which a classifier is trained on in vivo experimentally validated target gene motif features including conservation, frequency, and motif match PSSM scores among eight species per gene locus. Putative TF target genes are rank ordered by classifier label probabilities for a binary label (target gene or non-target gene).

## Training and validation using motif feature data

Motif feature data (diagramed in *Figure 3A*) from three *bona fide* terminal selector TF motifs with extensive in vivo validated TF target gene sets: the COE motif (UNC-3 binding site) (*Kratsios et al., 2011*); ASE motif (CHE-1 binding site) (*Etchberger et al., 2007*); AIY motif (CEH-10/TTX-3 binding site) (*Wenick and Hobert, 2004*) having 55 (*Glenwinkel et al., 2014*; *Kratsios et al., 2011*), 19 (*Wenick and Hobert, 2004*), and 26 (*Etchberger et al., 2007*) target genes respectively were used to build the training and validation sets. The training data consists of known target gene motif features and random putative target gene motif features (*Figure 3B,C*). Each data point is a gene in the reference genome (*C. elegans*) and its affiliated motif match features including upstream and intron frequency, log-odds PSSM scores, and alignment-independent conservation among eight nematode species (*Glenwinkel et al., 2014*; *Figure 3A*). We demonstrated the utility of predictive modeling from motif feature data using a customized cross-validation procedure.

## Cross-validation procedure

Briefly, a $10 \times 2$-fold cross-validation scheme was used. For each TF motif, the set of validated target genes N(v) and an equal sized set of randomly drawn target genes N(r) from whole genome putative target genes is generated. This set of N(v) validated gene and N(r) random gene motif feature data is generated 10 times (10 different sets of random genes). For each of the 10 sets (size 2N), the data is randomly shuffled and split in half. One half is used to train a classifier, the other half is used as the validation set. The training half and validation half are then swapped in a twofold validation scheme (*Figure 3B*). This procedure was performed on each of the three terminal selector motif datasets as well as one combined set using motif feature data from all three.

## Feature selection

Motif features for training the classifier were chosen through recursive feature elimination using the same cross-validation procedure repeated with 13 different classifiers to find the most informative model and feature set. Features excluded were motif match position relative to the transcription start site of a coding genes and combinatorial motif features including all averaged species motif features (see *Glenwinkel et al., 2014* for details). These include motif match PSSM scores and motif match frequency in upstream and intronic regions across species. All features shown in *Figure 3A* are used by the classifier to rank order candidate target genes.

## Classifier selection for identification of TF target genes

This training data is unique for a binary classification problem in that we have no true negative TF-independent genes. Instead we rely on a set of true positive TF target genes and use randomized data consisting of putative TF target genes with motif matches from the entire genome. In reality, some of these genes will indeed be true positives (real TF target genes), but we expect motif features among a randomized set have distinct features (such as lower conservation and PSSM scores, etc.) from a complete set of true positive TF target genes. Because we only know when true positives are correctly labeled as TF target genes and when false negatives are incorrectly labeled as non-target genes, we use recall (true positives/[true positives + false negatives]) to assess each classifier's performance. Additionally, we can test whether the rank order of correctly identified true positives in the validation set is significantly better than randomly drawn gene motif feature data using the Wilcoxon rank sums test. We found that a GPC modeled on the combined motif data from all three terminal selectors is the most accurate for the classification problem at hand (median recall = 0.8, median rank sums test Z score = 6.06, p<1e-9) with the most informative features being upstream and intronic motif match frequency, max log-odds motif score in upstream and intronic regions, plus upstream and intron motif match conservation among the eight nematode genomes (*Figure 3—figure supplement 1*). Given the success of the GPC classifier, we implemented a model built from the combined motif match data (100 true positive target gene motif match features + 100 random gene motif match features) into the TargetOrtho2 pipeline for rank ordering of putative TF target genes.

### Rationale for extending the classifier to other TF motif data

Examination of putative terminal selectors with available genetic loss-of-function data (*Supplementary file 2A*: genetic markers) shows that with increasing number of TF-dependent markers identified (and hence increasing confidence that the TF is a terminal selector), the probability of identifying motif enrichment and significant ranking of neuron class-specific genes from Wormbase also increases (*Figure 4—figure supplement 1*, *Figure 6—figure supplement 1*) suggesting that the GPC classifier is appropriately identifying target genes from novel TF motifs. Examination of another well-characterized terminal selector, HLH-4 that regulates ADL identity (*Masoudi et al., 2018*), shows significant enrichment and ranking of previously characterized *hlh-4* target genes in ADL (*Figure 4*) further supporting the use of the GPC trained on three terminal selector's motif feature data.

### Availability

The TargetOrtho2 software is available for download for OSX and Linux (see hobertlab.org/TargetOrtho2) (*Figure 3C*). Genome versions and sources are available with the software download.

### Parameters for running TargetOrtho2 with 136 DNA binding motifs

TargetOrtho2 was run using with a p-value threshold of 1e-4 for the FIMO scanner (option -p 1e-4), with the full intergenic regions and introns searched (option -d None) and with *C. elegans* as the reference genome (option -r C.elegans), scanning all eight included nematode species (option -s species.txt where all eight species are listed in the text file). The TargetOrtho2 data used in this analysis can be regenerated using a Meme formatted motif file (see http://meme-suite.org/) and the TargetOrtho2 software available at Hobertlab.org/TargetOrtho2.

## Data sources for binding site enrichment analysis

### Molecular markers of neuronal identity (=neuron type-specific gene batteries)

We previously extracted and manually curated gene expression data from Wormbase into an expression matrix that we called 'Brain Atlas' (*Hobert et al., 2016*). This data is based on published reporter gene expression profiles and is a representative readout of the regulatory state of each neuron type. Since then, additional expression patterns have become available and we updated Brain Atlas. Our latest version contains data for all 118 neuron classes, with each neuron class expressing on average 50 reporter genes (range 14 to 233).

### Single cell RNA sequencing profiles for 118 neuron classes

Aided by the molecular markers of neuronal identity from Brain Atlas, *Taylor et al., 2019* identified 118 neuron class clusters from scRNA of FACS sorted neurons from L4 animals. The VD and DD neuron classes did not separate into distinct clusters. These are analyzed together as one VD_DD cluster in the analysis. Single neuron expression profiling data provides an extensive set of differentially expressed markers for each neuron type (see differential expression analysis below) (*Figure 2A*).

### Neuronally expressed TFs with known DNA recognition motifs

PBM technology has been used to experimentally determine the DNA binding motifs of hundreds of *C. elegans* neuronal TFs (*Weirauch et al., 2014*). Here, we further investigate 136 neuronally expressed TFs with sufficiently unique DNA binding motifs (see motif curation procedure below) as assessed by similarity regression (SR) scores comparing *C. elegans* TF DNA binding domains (*Lambert et al., 2019*) (see Materials and methods). Four other motifs analyzed here were defined and validated through a combination of in vitro and in vivo studies (UNC-3, CEH-10/TTX-3, CHE-1, HLH-4 DNA binding motifs) (*Etchberger et al., 2007*; *Kratsios et al., 2011*; *Masoudi et al., 2018*; *Wenick and Hobert, 2004*; *Figure 2D*, *Supplementary file 1A*).

### DNA binding motif curation procedure

Starting from the set of differentially expressed TFs from single cell profiling and Wormbase reporters expressed in the nervous system (Brain Atlas, *Supplementary file 1C*), we identified 389

neuronal TFs. Using the CISBP database version 2.00 (*Weirauch et al., 2014*) to identify the best available motif from direct experimental evidence (or from the best inferred motif if no direct experimental motif exists), 214 neuronal TFs have available DNA binding motifs. We further removed motifs from TFs with highly similar DNA binding domains compared to other *C. elegans* TFs using the SR scores calculated in *Lambert et al., 2019* leaving 136 neuronal TFs with an available motif. We used an SR score cutoff of 0.807, the lowest score that allowed us to include *unc-42*, a homeodomain TF in our analysis which has a DNA binding domain most similar to *ceh-17* (SR score = 0.807). We also excluded duplicates where the same motif was inferred based on homology and cases where the best inferred motif was another *C. elegans* TF. For a full list of neuronal TFs and motifs including explanations for each exclusion, see (*Supplementary file 1A*).

## Differential expression analysis of scRNA data from CENGEN

UMI count data provided by *Taylor et al., 2021* for neuron type clusters were normalized using the SCTransform function in the Seurat 3.0 package (*Butler et al., 2018*; *Stuart et al., 2019*) followed by differential expression analysis using the findMarkers function with the Wilcoxon test (log-fold change threshold = 0, adjusted p-value threshold < 0.05). Neuron subclass clusters were combined prior to the differential expression analysis (ASEL and ASER combined into ASE, IL2_DV and IL2_LR combined into IL2, etc.) then each neuron class was tested against all other neuron classes (see *Figure 2* for a full list of neuron classes analyzed).

## Identifying motif signatures in neuron class gene batteries

Motif analysis was performed for each TF with an unambiguous motif (*Supplementary file 1A*, exclusion_rule = None) in each neuron class gene battery where the TF is expressed. This includes all TFs with reporter expression data in Brain Atlas plus those present in the CENGEN scRNA differential expression profiles per neuron class. Two tests were performed for each motif in each expressing neuron class gene battery: (a) hypergeometric test for motif enrichment and (b) Wilcoxon rank sums test. The enrichment test was performed using python's scipy.stats library's hypergeom function (scipy.stats.hypergeom.sf) on TargetOrtho2 data where the observed value is the number of neuron class gene loci with at least one motif match in upstream or intronic regions and the expected value is the number of coding genes in the genome with upstream or intronic motif matches divided by the total coding genes in the genome. The Wilcoxon rank sums test was performed using scipy.stats.ranksums where the TargetOrtho2 ranking of candidate TF target genes was compared between neuron class genes with motifs versus gene rankings across the *C. elegans* genome found by TargetOrtho2. For example, TargetOrtho2 identifies thousands of genes with upstream and/or intronic binding site matches. These candidate TF target genes are then rank ordered using the supervised learning algorithm described above. The Wilcoxon rank sums test compares the rank order of randomly drawn genes from the entire genome to the specific rank order of genes present in the neuron class battery of interest.

Using these two tests, we categorize motif signatures into three types, coordinated regulators, piecemeal regulators, and other. 'Coordinated regulator' is the strongest predictor of terminal selector regulation in which a motif is both significantly enriched and ranked among neuron class gene loci. Piecemeal regulators include TFs whose binding sites are not enriched in a specific neuron class gene battery, but the genes with a motif match present are significantly rank ordered by TargetOrtho2 compared to random genes in the genome with motif matches. The 'other' category includes less likely regulators in which the TF's binding sites are either not enriched in the neuron class gene battery or the neuron class genes with DNA binding motif matches are not significantly rank ordered by TargetOrtho2 as probable TF target genes compared to random genes with motif matches. DNA binding motif signatures for each regulator type are diagramed in (*Figure 4B*). p-Values were corrected for multiple testing using the Python statsmodels library multipletests function (method Benjamini/Hochberg, alpha = 0.05). See *Supplementary file 2* for all results.

## Cofactor motif analysis

Cofactor motif analysis was performed pairwise on all TF expressed in each neuron class. The hypergeometric test was performed as described above except the observed counts are the number of neuron class gene loci containing both TF motif matches in upstream and/or intronic regions. The

expected proportion is calculated from the proportion of gene loci with motif 1 across the genome from TargetOrtho2 multiplied by the proportion of gene loci with motif 2 across the genome. The Wilcoxon rank sums tests were performed as described above except the ranks of neuron class gene loci containing both motifs were compared to ranks of gene loci across the genome from TargetOrtho2 independently for each motif. If both motifs have significant ranks (p<0.05) among the genes with both motifs present, the test is considered significant (asterisk displayed in *Figure 7A*). Cofactor motif signature models are diagrammed in *Figure 7A*. We observe three distinct co-motif signatures diagramed in *Figure 7A*: (a) 'co-occurring coordinated signatures' (*Figure 7A*, orange) in which not only is each TF individually characterized as a 'coordinated regulator' (*Figure 4A*, blue), but the observed DNA binding motif matches from each of two TFs are significantly co-enriched in common neuron class-specific genes compared to expected by chance (*Figure 7A*, co-enrichment test p<0.05) and these common genes are each significantly rank ordered by TargetOrtho2 compared to random genes with DNA binding motif matches in the genome (*Figure 7A*, gene rank tests, p<0.05 for each motif); (b) 'co-occurrence of piecemeal signatures', in which significant co-enrichment and TargetOrtho2 rank order is observed as in case 1, but one or both of the two TFs may be a piecemeal regulator (*Figure 7A*, yellow); (c) 'separate targets', in which either significant co-enrichment and/or TargetOrtho2 rank order is not observed for one or both motifs (*Figure 7A*, gray). p-Values were corrected for multiple testing using the Python statsmodels library multipletests function (method Benjamini/Hochberg, alpha = 0.05). See *Supplementary file 2D* for all pairwise results.

## Examination of 4 *bona fide* terminal selector DNA binding site enrichments using TargetOrtho2

As a first application of TargetOrtho2, we examined differential neuronal gene expression from scRNA sequencing data from four terminal selector TFs with in vivo evidence of neuronal identity gene regulation: (UNC-3 in DA, DB, VA, VB; CHE-1 in ASE; TTX-3 in AIY; and HLH-4 in ADL), each having more than 20 reporter genes examined in a genetic loss-of-function mutant (*Figure 1B,C*). We were able to examine entire differential expression profiles of the neuron types regulated by these four factors (UNC-3 regulates DA, DB, VA, VB; CHE-1 regulates ASE; TTX-3 regulates AIY; and HLH-4 regulates ADL identity) (*Etchberger et al., 2007*; *Kratsios et al., 2011*; *Masoudi et al., 2018*; *Wenick and Hobert, 2004*). We found that TF binding sites for these four terminal selectors are enriched and significantly rank ordered as candidate target genes by TargetOrtho2 (*Figure 4C*, *Supplementary file 2B*). Previous work from *Wenick and Hobert, 2004* identified an AIY motif that is bound by a TTX-3::CEH-10 heterodimer. This motif was useful in identifying novel AIY expressed reporter genes and mutation analysis showed a requirement of the motif for expression. Furthermore, 17/20 AIY markers are turned off in the *ttx-3* mutant (*Wenick and Hobert, 2004*; *Figure 1C*). Here, we confirm that both TTX-3 and CEH-10 predicted that binding sites are independently enriched and significantly ranked by TargetOrtho2 among the 177 AIY differentially expressed genes supporting a terminal selector cofactor role. Clustering of AIY genes by TargetOrtho2 rank allows comparison of the in vivo derived AIY motif from *Wenick and Hobert, 2004* with the two in vitro derived CEH-10 and TTX-3 DNA binding motifs from PBM experiments (*Barrera et al., 2016*; *Weirauch et al., 2014*). As expected, we observe extensive overlap in AIY genes among the three AIY motifs suggesting in vitro and in vivo methods result in congruent DNA binding site representations (*Figure 4C*).

## Comparing TargetOrtho2 motif enrichments from in vivo validated motifs to in vitro derived DNA binding motifs

TargetOrtho2 was trained on in vivo validated target genes from three of these terminal selectors with experimentally derived motifs: UNC-3, CHE-1, and CEH-10::TTX-3 (DNA binding motifs from *Etchberger et al., 2007*; *Kratsios et al., 2011*; *Wenick and Hobert, 2004*). To demonstrate that motif enrichment and ranking of target genes by TargetOrtho2 was not just as artifact of training the pipeline on motif features derived from these specific motifs, we also examined each target neuron class gene battery (*unc-3*:DA/DB/VA/VB, *che-1*:ASE, *ceh-10*/*ttx-3*:AIY) using alternative motifs independently derived from in vitro binding assays (PBM motifs from *Barrera et al., 2016*; *Narasimhan et al., 2015*; *Nitta et al., 2015*; *Weirauch et al., 2014*). We found that binding site enrichments and TargetOrtho2 rank order were also significant with highly correlated TargetOrtho2

rank percentiles per neuron class gene battery (Spearman correlation p<<1e-3 for each comparison) (*Figure 3—figure supplement 3A*). We undertook the same analysis using the manually curated, Wormbase-extracted reporter gene expression atlas (termed 'Brain Atlas'). Similar binding site enrichments were detected with this dataset (*Figure 3B*, *Figure 3—figure supplement 3B*). This finding suggests that four well-studied terminal selectors likely exert their coordinated regulatory function through direct *cis*-regulation of neuron class-specific effector genes.

## Transgenic line generation and in vivo expression analysis

GFP promoter fusions were generated as in *Hobert, 2002* for *ins-1* and *pgp-2* loci using the following primers:

> ins-1. Primer A*: cgaacgtgcttctcagcata
> Primer C: GAAAAGTTCTTCTCCTTTACTCATaacttgacgaaaccagtacat.
> Primer D*: CTCTGACACATGCAGCTCCC,
> pgp-2 promoter including exon1 and intron.
> Primer A*: caagaatgcagttcggtgag.
> Primer C: GAAAAGTTCTTCTCCTTTACTCATgaagagtcacttgaatgagag.

Transgenic lines were microinjected into hermaphrodites by the Hobert lab technician Qi Chen. Animals were imaged on a confocal microscope at 63×. Strain information is available in *Supplementary file 4*.

## Binding site deletions

Restriction free cloning (*van den Ent and Löwe, 2006*) was used to generate binding site deletions for *unc-42* and *lim-6* in the *cho-1* promoter using the following primers:

> *lim-4/unc-42* binding deletion:
> 5'-tccttcaacaaataaaattcaaaaataaattatctcacaaagatttatcatttctggaggag,
> 5'-ctcctccagaaatgataaatctttgtgagataatttattttttgaattttatttgttgaagga.
> *unc-42* binding site deletion:
> 5'-cttctcttaaaaagaaggttgtctttcccctattcatttttccatg,
> 5'-catggaaaaatgaatagggggaaagacaaccttcttttttaagagaag.

The unc-86 binding site (taaataatta) was deleted using a two-piece Gibson Assembly that introduced the mutation with primers.

# Additional information

## Competing interests

Oliver Hobert: Reviewing editor, *eLife*. The other authors declare that no competing interests exist.

## Funding

| Funder | Grant reference number | Author |
| --- | --- | --- |
| Howard Hughes Medical Institute | | Oliver Hobert |
| National Institutes of Health | R01NS100547 | David M Miller III<br>Marc Hammarlund<br>Oliver Hobert<br>Nenad Sestan |

The funders had no role in study design, data collection and interpretation, or the decision to submit the work for publication.

## Author contributions

Lori Glenwinkel, Conceptualization, Resources, Data curation, Software, Formal analysis, Validation, Investigation, Visualization, Methodology, Writing - original draft; Seth R Taylor, Conceptualization, Data curation, Formal analysis, Investigation, Methodology, Writing - review and editing; Kasper

Langebeck-Jensen, Laura Pereira, Molly B Reilly, Manasa Basavaraju, Ibnul Rafi, Eviatar Yemini, Formal analysis, Investigation; Roger Pocock, Nenad Sestan, Supervision, Project administration; Marc Hammarlund, Supervision, Funding acquisition, Project administration, Writing - review and editing; David M Miller III, Conceptualization, Supervision, Project administration, Writing - review and editing; Oliver Hobert, Conceptualization, Funding acquisition, Writing - original draft, Project administration

## Author ORCIDs
Lori Glenwinkel (iD) https://orcid.org/0000-0003-4874-6146
Molly B Reilly (iD) http://orcid.org/0000-0002-7180-7763
Roger Pocock (iD) http://orcid.org/0000-0002-5515-3608
Nenad Sestan (iD) http://orcid.org/0000-0003-0966-9619
David M Miller III (iD) http://orcid.org/0000-0001-9048-873X
Oliver Hobert (iD) https://orcid.org/0000-0002-7634-2854

## Decision letter and Author response
Decision letter https://doi.org/10.7554/eLife.64906.sa1
Author response https://doi.org/10.7554/eLife.64906.sa2

## Additional files

### Supplementary files
• Supplementary file 1. Data sources. (A) Motif information. Log-odd PSSM in MEME format for non-CISBP source motifs can be found at the end of (A). (B) Single cell differentially expressed genes as binary expression matrix. (C) Brain Atlas binary expression matrix.

• Supplementary file 2. Motif analysis results from single cell differential expression data. (A) Results from transcription factor (TF) with genetic loss-of-function data including results that were further validated in vivo. (B) Results for four, well-characterized terminal selectors. (C) Nervous system-wide motif analysis results from single cell differential gene expression data. (D) Cofactor results from motif analysis of single cell differential gene expression data. (E) All predicted coordinated regulators (candidate terminal selectors) by neuron class. (F) Candidate target neuron class gene batteries by coordinated regulatory TF. (G) All co-occuring coordinated signatures (candidate terminal selector co-regulators).

• Supplementary file 3. Motif analysis stats from Brain Atlas reporter data. (A) Brain Atlas single factor motif analysis results. (B) Cofactor results. (C) Agreement of top two regulators between datasets.

• Supplementary file 4. Strain information.

• Transparent reporting form

### Data availability
TargetOrtho2 software code and other code used in this study are available as a git repository at https://github.com/loriglenwinkel/TargetOrtho2.0 (copy archived at https://archive.softwareheritage.org/swh:1:rev:f69d7516374f84c67c37a4bbefed3995312889f4). Single cell differential expression profiles per neuron class, Wormbase neuronal reporter genes (Brain Atlas), and TF binding site enrichment results among neuron class gene batteries can be found in the supplemental materials.

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
