## [Decision Letter]

**Acceptance summary:**

This is a tour de force manuscript in which the authors comprehensively analyze all worm neurons for single cell RNA sequence data, transgene expression data, and conserved transcription factor binding site data. They use this massive dataset to test the hypothesis that distinct single or pairs of transcription factors specify the identity of all 118 known neuronal classes. The manuscript beautifully integrates previous datasets and provides some new experimental tests of their proposed terminal selector model.

**Decision letter after peer review:**

Thank you for submitting your article "Transcriptional regulatory logic of neuronal identity specification throughout the *C. elegans* nervous system" for consideration by *eLife*. Your article has been reviewed by 2 peer reviewers, one of whom is a member of our Board of Reviewing Editors, and the evaluation has been overseen by Piali Sengupta as the Senior Editor. The reviewers have opted to remain anonymous.

The reviewers have discussed the reviews with one another and the Reviewing Editor has drafted this decision to help you prepare a revised submission.

Summary:

This study combines single cell RNA-seq and transcription factor DNA-binding data to help resolve the regulatory logic behind neuronal sub-type specification in *C. elegans*. They find that most neurons appear to be specified not by one TF but by a combination of TFs. They also find that while the majority of TFs act on the whole cohort of effector genes that define each neuron type, a minority of TFs only appear to act on a subset of effector genes. That some TFs play a more comprehensive role while others only control sub-routines has also been demonstrated before. Hence, the study is not conceptually novel. Yet, the study represents a major leap forward due to the complete identification of all neuronal sub-classes in C.elegans, and because the compressed worm genome allows for meaningful TF-to-target-gene predictions, something that is difficult in mammals. From these perspectives the study will be of broad interest.

Essential revisions:

The authors comprehensively analyze all worm neurons for scRNA sequence data, transgene expression data, and conserved TF binding site data. They test the hypothesis that distinct single or pairs of TFs act as terminal selectors for each of the 118 defined neuronal classes. This is in contrast to "piecemeal" regulation of each neuronal class by multiple TFs that each control a small portion of the relevant gene expression program. The manuscript integrates previous data sets and provides some new experimental tests of their proposed terminal selector model. They find that most neurons appear to be specified not by one TF but by a combination of TFs. This concept was previously demonstrated in flies and mice, but only for selected neurons. Here they find that while the majority of TFs act on the whole cohort of effector genes that define each neuron type, a minority of TFs act on only a subset of effector genes. However, they use conflicting terminology for these important TFs that should be clarified for the benefit of the wider neural development field, as *C. elegans* often leads the way for other model systems. Although the study is not conceptually novel, it represents a major leap forward due to the complete identification of all neuronal sub-classes in C.elegans, and because the compressed worm genome allows for meaningful TF-to-target-gene predictions, something that is difficult in mammals. Finally, the paper generates interesting testable hypotheses for understanding neuronal diversity in other animals.

The majority deal with presenting a clear definition of terms used.

1. As the authors point out, the linkage of TFs to target genes based upon known in vitro or in silico predicted DNA binding is merely correlative, and finding target sites in a downstream gene frustratingly often does not translate into direct regulation. There are several reasons for the limited correlation between TF sites and direct regulation e.g. change in TF PWM when binding as monomer or homodimer, change in TF PWM when binding as monomer or heterodimer, and because bona fide target sites can be low affinity sites that do not match the PWM (PMID 25557079). Yet the Abstract states that "our nervous system wide analysis at single cell resolution demonstrates that many transcription factors directly co-regulate the cohort of effector genes that define a neuron type". This phrasing implies causal relationships rather than correlative findings. Similarly, explicitly stating that some TFs are "piece meal" regulators while others are "master regulators" chiefly based upon correlative data also seems to be a bit of a stretch. It would be a good hypothesis to leave for future functional studies.

2. The term "master regulator" was introduced already in the 80s and early 90s, triggered by the potency of e.g., mammalian MyoD to convert fibroblasts into myoblasts, by Antp to convert the fly antennae into legs, and by Eya (Pax6) in converting fly legs into eyes. The authors previously deserted this nomenclature, and introduced the name "terminal selectors", defined as TFs that (A) directly regulate (B) all aspects of a neuron's identity, and (C) acted throughout the neuron's life.

However, because many TFs do not fit this description (indeed, there is probably not a single fly or mouse TF that currently qualifies on all three counts) they are now running into "semantical problems". Specifically:

A. In this study, indirect regulation by some TFs on some target gene(s) is as high as 36%. By definition (A) these TFs are not "terminal selectors" then. What should these TFs be called? "Indirectly acting terminal selectors"?

B1. Some of the regulation is "piece meal" i.e., some TFs only control expression of a subset of the sub-type specific genes. What should such "piece meal" "terminal selectors" be called? They cannot be called "terminal selectors", based upon the original definition (B). They are likely to be very common in flies and mammals.

B2. They re-introduce the name "master regulator". But by their original definition a "terminal selector" is already a "master regulator" i.e., it controls all aspects of a neuron's identity (B). So what is the difference between a "master regulator" and a "terminal selector"?

C. It is unclear if all of the TFs studied herein are indeed expressed throughout the worm's life. If not, they do not qualify as "terminal selectors" according to the author's definition (C). This semantic problem applies even more to more derived animals, and the vast majority of fly and mammalian TFs governing neuronal sub-type cell fate during development are not maintained in the same neurons in the adult CNS (Allan Brain Atlas, literature). So the criteria that a "terminal selector" should be expressed throughout a neuron's life would also disqualify the vast majority of fly and mammalian TFs from being "terminal selectors".

These semantic issues could perhaps be resolved by applying a broader definition of terminal selectors as: any TF that determines any unique aspect of cell fate, acting transiently or throughout the cell's life, and directly or indirectly.

3A. it is important to define common terms in the introduction on first use: identity, neuron class, neuron cell type, and gene battery. For example, cholinergic neurons do not fall into any of these categories despite NT identity being a commonly used marker of cell type/neuron identity.

3B. Are neurons in a 'class' molecularly and morphologically identical? Are they left/right pairs? It would be good to know not only what makes a 'neuronal class' but what differences may exist within a class.

4. Figure 4A,B shows many neuron class specific genes with <50% containing a DNA-binding motif. How are these 'motif-less' genes properly expressed without input from the terminal selector TF? Indirectly? If so, it wasn't clearly stated what all low motif genes were indirectly regulated. Or are they regulated by a different TF with a higher percentage of motifs/gene? Or are there simply about half the motifs that are too degenerate to be counted by the program, yet still functional?

5. Many terminal selector genes are widely expressed in multiple neuron classes: unc-86, egl-13, unc-3, ceh-43 (Figure 4A). Do these terminal selectors always function as heterodimers? Are their partners shown somewhere? That was not clear.

6. The introduction is a masterful summary; in contrast, the discussion is very limited. Discussion is short, poorly referenced and does not extend beyond *C. elegans*. It would be an improvement to discuss existing data from other systems that supports or conflicts with your terminal selector model, particularly in *Drosophila*, mouse, and perhaps chick spinal cord.

7. It would be nice to include a few legible views of the PWM images for binding sites used. Particularly since homeodomain binding sites have such low information content. I think there were one or two in one of the figures, but it was too small to read. Perhaps as a supplemental figure?

– Line 447 name and define the common properties for "command neurons" on first use.

– Typo line 344? says 117 not 118.

– Line 123 how many total TFs beyond 394? How many TFs in CNS beyond 136? I eventually found them in microscopic type in Figure 2B, but please add them to the main text. E.g. 395/~900 and 136/419.

– Can TargetOrtho2 be adapted to run on any critter with a sequenced genome in multiple species, such as *Drosophila*? Can TargetOrtho2 be used for de novo binding site discovery? Given a TF lacking a known binding motif, expressed in a neuron where there are >50 CRM-reporter transgenes with + or – expression in the neuron, can the program identify over-represented motifs that would be putative TF binding sites? These things could be added to the discussion.

– Consider moving the paragraph on lines 353-371 to the Discussion.

– The figures are far too small to read when printed out; I'll leave it to the editors to decide on whether e-viewing alone is necessary for a figure, or does it need to be legible in print form.

---

## [Author Response]

Essential revisions:*C. elegans*[…] The majority deal with presenting a clear definition of terms used.1. As the authors point out, the linkage of TFs to target genes based upon known in vitro or in silico predicted DNA binding is merely correlative, and finding target sites in a downstream gene frustratingly often does not translate into direct regulation. There are several reasons for the limited correlation between TF sites and direct regulation e.g. change in TF PWM when binding as monomer or homodimer, change in TF PWM when binding as monomer or heterodimer, and because bona fide target sites can be low affinity sites that do not match the PWM (PMID 25557079). Yet the Abstract states that "our nervous system wide analysis at single cell resolution demonstrates that many transcription factors directly co-regulate the cohort of effector genes that define a neuron type". This phrasing implies causal relationships rather than correlative findings. Similarly, explicitly stating that some TFs are "piece meal" regulators while others are "master regulators" chiefly based upon correlative data also seems to be a bit of a stretch. It would be a good hypothesis to leave for future functional studies.

In the Abstract, we changed “demonstrates” to “indicates”. Nevertheless, we wish to point out that our analysis is not purely correlational, i.e. our conclusions are not merely based on presence/absence of binding sites. There is a very substantial amount of reporter analysis done – mainly in the literature, but also in this manuscript – that shows that shows that many of the TFs under investigation have an effect on reporter gene expression, and we show here that this genetic dependency is parallel by the presence of binding sites. In several cases, we have also mutated such binding sites.

2. The term "master regulator" was introduced already in the 80s and early 90s, triggered by the potency of e.g., mammalian MyoD to convert fibroblasts into myoblasts, by Antp to convert the fly antennae into legs, and by Eya (Pax6) in converting fly legs into eyes. The authors previously deserted this nomenclature, and introduced the name "terminal selectors", defined as TFs that (A) directly regulate (B) all aspects of a neuron's identity, and (C) acted throughout the neuron's life.However, because many TFs do not fit this description (indeed, there is probably not a single fly or mouse TF that currently qualifies on all three counts) they are now running into "semantical problems". Specifically:A. In this study, indirect regulation by some TFs on some target gene(s) is as high as 36%. By definition (A) these TFs are not "terminal selectors" then. What should these TFs be called? "Indirectly acting terminal selectors"?

We very much appreciate that these definitional issues are brought up – it has helped us to improve terminology throughout the manuscript. It is true that direct regulation (point A) has been a feature of the terminal selector definition. However: absence of evidence is not evidence of absence. If we see a good site that is conserved we infer it’s functional. But if we do NOT see a good site that is conserved, we can’t exclude the terminal selector does not bind: it may bind to a poorer site that’s below our detection threshold, or the site may simply not be conserved in other nematode species. Hence, because the absence of direct regulation is difficult to prove, perhaps it is more useful for the definition of a terminal selector to “only” invoke the operational, genetic criteria of a neuron’s identity being very severely affected, be that via direct regulation or not. This is of course along the lines of the original definition “selector”, which also made no assumptions about biochemical mechanisms, but rather invoked the operational definition of affecting the identity of developing fields/organs. The “terminal” addition to “selector” – an important extension of this concept – should be viewed as operational as well, in the sense that terminal identity features are gone in the mutant, without affecting the generation of the neuron. Plus, terminal identity features are continuously controlled. This is often direct, but may not have to be.

In the revised manuscript, we now discuss this point, we hope, clearly and succinctly, in a substantially revised Discussion (which was requested to be improved anyway, so that fits well). We have also carefully combed through the manuscript and minimized the usage of the word “master regulatory” and now instead call such regulation “coordinated regulatory control”, to indicate that one factor (or a specific factor combination) coordinate the expression of many members of a gene battery.

B1. Some of the regulation is "piece meal" i.e., some TFs only control expression of a subset of the sub-type specific genes. What should such "piece meal" "terminal selectors" be called? They cannot be called "terminal selectors", based upon the original definition (B). They are likely to be very common in flies and mammals.B2. They re-introduce the name "master regulator". But by their original definition a "terminal selector" is already a "master regulator" i.e., it controls all aspects of a neuron's identity (B). So what is the difference between a "master regulator" and a "terminal selector"?

This is again a spot-on criticism that indeed requires clarification. First and foremost, we indeed do NOT believe that piecemeal regulators should be called terminal selectors; that would not properly reflect the intellectual history of the “selector” terminology. The reason we “revived” here the “master regulatory” term is to contrast the “terminal selector” mode of regulation with the “piece-meal” regulation role. Hence, in the original version of the manuscript, we appended the term “master regulatory” to “terminal selector”, hence making it “Piecemeal TFs” vs “master-regulatory terminal selectors”. With the reviewer’s criticism we are now realizing that we’re wading into old nomenclature issues about what “master regulatory” traditionally means. It used to mean (based on factors like MyoD or Pax6/eyeless) that such factors have the ability of factors to ECTOPICALLY induce specific fates. Unfortunately, the term has become loaded with this sufficiency criterion, which never was even true – because all the above-mentioned factors were only able to exert their “master regulatory” function in specific cellular contexts. To avoid moving into such terminology territory, we minimized the usage of the word “master regulatory” and now instead call such regulation “coordinated regulatory control”, to indicate that one factor (or a specific factor combination) coordinate the expression of many members of a gene battery.

C. It is unclear if all of the TFs studied herein are indeed expressed throughout the worm's life. If not, they do not qualify as "terminal selectors" according to the author's definition (C). This semantic problem applies even more to more derived animals, and the vast majority of fly and mammalian TFs governing neuronal sub-type cell fate during development are not maintained in the same neurons in the adult CNS (Allan Brain Atlas, literature). So the criteria that a "terminal selector" should be expressed throughout a neuron's life would also disqualify the vast majority of fly and mammalian TFs from being "terminal selectors".These semantic issues could perhaps be resolved by applying a broader definition of terminal selectors as: any TF that determines any unique aspect of cell fate, acting transiently or throughout the cell's life, and directly or indirectly.

All of the TFs studied in this paper are expressed throughout the worm’s life – and the reviewer is totally correct to point out that this is a critical criterion for being called a terminal selector. We have now clarified this in the Introduction and Discussion.

3A. it is important to define common terms in the introduction on first use: identity, neuron class, neuron cell type, and gene battery. For example, cholinergic neurons do not fall into any of these categories despite NT identity being a commonly used marker of cell type/neuron identity.

We are not entirely sure that we understand the example that the reviewers brings up. Cholinergic markers are undoubtedly a key feature of neuron type-specific gene batteries and, hence, a key feature of neuronal identity. But, of course, they are NOT the sole defining feature of a neuron’s identity; many different types of neurons are cholinergic. But this is a feature for most molecular identity markers are a neuron – they are rarely ever exclusively expressed in one neuron class and nowhere else. Perhaps this is where the request for definition of identity/neuron class/type etc. comes from? We thought we explained this well enough in the first paragraph of the Introduction, but we now appended this paragraph to make this clearer: Any neuron-type specific gene batteries that defines a neuron’s identity is composed of individual features that are shared by other neurons as well.

3B. Are neurons in a 'class' molecularly and morphologically identical? Are they left/right pairs? It would be good to know not only what makes a 'neuronal class' but what differences may exist within a class.

Classes were morphologically defined originally. Mostly they are bilaterally symmetric pairs of neurons. There can indeed be subtle anatomical (and molecular) differences between individual class members. Our analysis does not touch on this subject, so we thought to not go into this in any detail but to just state that there are 118 anatomically defined classes.

4. Figure 4A,B shows many neuron class specific genes with <50% containing a DNA-binding motif. How are these 'motif-less' genes properly expressed without input from the terminal selector TF? Indirectly? If so, it wasn't clearly stated what all low motif genes were indirectly regulated. Or are they regulated by a different TF with a higher percentage of motifs/gene? Or are there simply about half the motifs that are too degenerate to be counted by the program, yet still functional?

We know discuss the issue of possible indirect regulation and/or additional regulator in the Discussion.

5. Many terminal selector genes are widely expressed in multiple neuron classes: unc-86, egl-13, unc-3, ceh-43 (Figure 4A). Do these terminal selectors always function as heterodimers? Are their partners shown somewhere? That was not clear.

Yes, they usually act in different combination in different cell types. Discussed now in text.

6. The introduction is a masterful summary; in contrast, the discussion is very limited. Discussion is short, poorly referenced and does not extend beyond *C. elegans*. It would be an improvement to discuss existing data from other systems that supports or conflicts with your terminal selector model, particularly in *Drosophila*, mouse, and perhaps chick spinal cord.

We have extended the Discussion, with point already mentioned above, but also with Discussion on the potential of conservation of terminal selectors in other organisms.

7. It would be nice to include a few legible views of the PWM images for binding sites used. Particularly since homeodomain binding sites have such low information content. I think there were one or two in one of the figures, but it was too small to read. Perhaps as a supplemental figure?

Yes, added as new Supp. Figure.

– Line 447 name and define the common properties for "command neurons" on first use.

Fixed.

– Typo line 344? says 117 not 118.

117 of 118 neuron class were found to have a coordinated regulator. We updated this in the manuscript.

– Line 123 how many total TFs beyond 394? How many TFs in CNS beyond 136? I eventually found them in microscopic type in Figure 2B, but please add them to the main text. E.g. 395/~900 and 136/419.

We updated the manuscript to include these details.

– Can TargetOrtho2 be adapted to run on any critter with a sequenced genome in multiple species, such as *Drosophila*? Can TargetOrtho2 be used for de novo binding site discovery? Given a TF lacking a known binding motif, expressed in a neuron where there are >50 CRM-reporter transgenes with + or – expression in the neuron, can the program identify over-represented motifs that would be putative TF binding sites? These things could be added to the discussion.

TargetOrtho is not designed for de novo motif extraction. Other tools have attempted to do this, with somewhat mixed success. We mention this now explicitly in the Discussion.

As per adaptation for other organisms: Yes, absolutely. We now mention this explicitly in the Discussion.

– Consider moving the paragraph on lines 353-371 to the Discussion.

This paragraph discusses the important point that not all genes in a neuron-type specific gene battery are necessarily direct regulators of a terminal selector – this point was brought up above already, it’s an obvious point that warrants description and discussion where we describe this result and we would therefore prefer to leave this here.